



# Preliminary evaluation of the effect of electro-coalescence with conducting sphere approximation on the formation of warm cumulus clouds using SCALE-SDM version 0.2.5-2.3.0

Ruyi Zhang[1], Limin Zhou[1,2], Shin-ichiro Shima[3,4], Huawei Yang[1]

[1]Key Laboratory of Geographic Information Science, Ministry of Education, East China Normal University, Shanghai, 200241, China
[2]Key Laboratory of Numerical Modeling for Atmospheric Science and Geophysical Fluid Dynamics, Institute of Atmospheric Physics, CAS, Beijing, 100029, China
[3]Graduate School of Simulation Studies, University of Hyogo, Kobe, 6512103, Japan
[4]RIKEN Center for Computational Science, Kobe, 6500047, Japan

*Correspondence to*: Limin Zhou(lmzhou@geo.ecnu.edu.cn)

**Abstract.** The analytic expression for electro-coalescence with the accurate electrostatic force for a
pair of droplets with opposite sign charges is established by treating the droplets as conducting spheres (CSs). Then, the weak electric effect on a cumulus cloud is investigated by size resolved cloud model with particle treatment of the super-droplet method. The results show that with CS treatment, the electrostatic force contributes a larger effect on cloud evolution than previous research. With a 3% lower charge limit of the maximum charge amount of the droplet, the domain total precipitation with CS
treatment for droplets with opposite signs is 52.5% higher than that with the no charge (NC) setting. Compared with previous work by Khain et al. (2004), with the multi-image-dipole treatment of CS, the amount of precipitation is 5.42% higher. It is found that the charged droplets could affect cloud formation even when the droplet charge is lower charge limit. High pollution levels result in greater sensitivity to electro-coalescence. The results show that when the charges ratio between two droplets is over 100, the
short-range attractive electric force due to the multi-image dipole would also significantly enhance precipitation for the cumulus. It is indicated that although the accurate treatment of the electrostatic force with CS method would require 30% longer computation time than before, it is worthwhile to include it in cloud, weather, and climate models.




## 1 Introduction

Clouds are considered to play a key role in climate systems, and the collision-coalescence of cloud droplets plays a key role in cloud formation. Droplet coalescence is one of the main processes leading to precipitation and even cloud chemistry, affecting cloud microphysics and thereby changing the global
radiation budget (Craig, 1995; Forbes and Clark, 2003; Rosenfeld et al., 2008). Several studies have reported that the electrostatics on charged droplets could significantly influence the droplet coalescence and droplet-aerosol coagulation in weakly electrified clouds (Pruppacher and Klett, 1997; Tinlsey et al., 2001, 2006; Zhou et al., 2009; Tripathi et al., 2008; Zhang et al., 2019). This electrostatic force induced effect is called electro-coalescence or electro-anti-coalescence (Tinsley, 2008) and could even explain
the link between solar wind fluctuations and changes in atmospheric parameters, such as cloud cover, polar surface pressure and the effective radiation in polar regions (Kniveton et al., 2008; Lam et al., 2014; Frederick et al., 2018, 2019).

In weakly electrified clouds, the accumulation of space charges on droplets is controlled by the diffusion of atmospheric ions produced by the cosmic ray flux, and the concentration is dependent on the ratio of
attachment and recombination and the downward ionosphere-earth current density ($J_z$). When the $J_z$ penetrates the cloud, the gradients of the electric field at the cloud boundary could generate net positively charged droplets at the upper cloud boundary and net negatively charged droplets at the lower boundary (Zhou and Tinsley, 2007; Nicoll and Harrison, 2016). The observations of Beard (2004) revealed that with a $J_z$ of 1-6 pAm$^{-2}$ in stratocumulus and altostratus clouds, a cloud droplet with radius of 10 $\mu m$ can
accept approximately 100 elemental charges, which is consistent with the theoretical calculations by Zhou and Tinsley (2007). In the cumulus, with vertical convection, the charged droplets at the boundaries can be mixed. The maximum charge on the droplets is determined by the air breakdown voltage for corona discharge (Meek and Carggs, 1953) and is a quadratic function of the droplet radius (Khain et al., 2004; Andronache, 2004).

In the 1970s, the collision efficiency of oppositely charged droplets evaluated with a centered Coulomb force indicated that only in strongly electrified clouds can the charge on droplets significantly affect cloud droplet coagulation (Wang et al., 1978). The series of trajectory simulation work by Tinsley et al. (2001, 2006), Zhou et al., (2009) and Tripathi et al., (2006) revealed that in a weakly electrified cloud, when taking into account the image charge force, the collision rate coefficient between the charged
droplets could be different. Even with droplet charges of the same sign, the collision rate coefficient could be enhanced as a function of the charge on the particles with radii ranging from 0.1 microns to 10 microns (Zhou et al., 2009). Then, with a sufficiently large charge on the droplets, the well-known Greenfield gap (Greenfield, 1957) could be closed. Simulation results showed that for particles with radii smaller than 0.1 microns, when the particles obtain a large charge due to the evaporation of highly
charged droplets, the collision rate coefficient is significantly decreased due to the repulsive electric force of droplets with charges of the same sign and is increased for charges of the opposite sign (Tinsley and Leddon, 2013). The updated simulation by Zhou et al. (2009) with an exact electric force treatment with the conducting sphere (CS) method indicated that the collision efficiency is a factor of two higher in the Greenfield Gap than that from the results of single image charge (IM) treatment. A few laboratory
experiment results were consistent with these theoretical simulations (Ardon-Dryer et al., 2015). These





findings highlight the need to represent coagulation due to droplet and aerosol charges in the cloud model. Khain et al. (2004) (hereafter Khain04) conducted a 0-dimension simulation to study the effect of seeding charged droplets on a cumulus cloud using the spectral bin cloud model with a 4-dimensional collision efficiency lookup table based on the static electric force between charged droplets. The results showed a

significant response in the evolution of clouds due to charged droplets. 5% of maximum charge amounts of natural droplets, which is 2.5 times larger than the results from Zhou et al. (2007), was used in Khain04 to investigate their influence on rain enhancement. Andronache et al. (2004) and Wang et al. (2015) claimed that charged droplets significantly contribute to below cloud scavenging according to the analytical formula suggested by Davenport and Peters (1978), where the minimum amount of charge on

droplets is 7% of the maximum limit. However, only Coulomb force (CB) treatment was used in Andronache's and Wang's simulation.

In this work, the effect of the electro-coalescence from $J_z$ on a warm cumulus with an exact electric force treatment with the CS method is estimated based on particle–based cloud modeling with the real-time collision efficiency calculation using the super-droplet method. The lower charging rate threshold for

electro-coalescence is discussed. The extreme assumption of the droplet charging scenario of opposite sign charge is investigated. The electro-anti-coalescence (Tinsley and Zhou (2015)) between charged droplets and particles could also be important for deep convection and stratus cloud evolution, and this will be addressed in future work.

## 2. Description of the cloud model

A particle-based cloud model is used with the particle size resolved treatment following the Super-Droplet Method (SDM) by Shima et al. (2009, 2020). This section provides an accurate description of the super-droplets method, how we generalize the exact electric force treatment with the CS method approach for the cloud model, and the numerical simulation setup.

### 2.1 Definition of super-droplets


Super-droplets have been defined in detail by Shima et al. (2009, 2020). A super-droplet represents multiple droplets with the same attributes and position, and this multiplicity is denoted by the positive integer $\xi_i(t)$, which can be different in each super-droplet and is time-dependent due to the definition of coalescence. Then, each super-droplet has its own position $x_i(t)$ and its own attributes $a_i(t)$ that

characterize the $\xi_i(t)$ identical droplets represented by super-droplet $i$. In this study, we assume that the attributes consist of the equivalent radius of water and the ammonium bisulfate mass in the droplet:: $a_i(t) = [R_i(t), M_i(t)]$. Since each real droplet takes different positions and attributes, a super-droplet is a kind of coarse-grained view of droplets both in real space and attribute space. Assume that $N_s(t)$ is the number of super-droplets floating in the atmosphere at time $t$. Then, the super-droplets represent $N_r(t) =$

$\sum_{i=1}^{N_s(t)} \xi_i(t)$ real droplets in total.



### 2.2 Motion of a super-droplet

The advection and sediments were described in detail by Shima et al. (2009, 2020) as follows:

$$\frac{d(m_i v_i)}{dt} = F_i^{drg} - m_i g \hat{z} \tag{1}$$

where $m_i = (4\pi/3)R_i^3 \rho_{liq}$ is the mass of droplet $i$, and $\rho_{liq} = 1.0 \ g \ cm^{-3}$ is the density of liquid water. $F_i^{drg} = m_i g \hat{z} + d(m_i v_i)/dt$ is the drag force from moist air, $g$ is the gravity of Earth, and $\hat{z}$ is the unit vector in the direction of the z-axis. $-F_i^{drg}$ gives the reaction force acting on the moist air. Considering that the relaxation to the terminal velocity is instantaneous, and the equation of motion becomes:

$$v_i = U_i - \hat{z} v_i^\infty, \ \ \frac{dx_i}{dt} = v_i \tag{2}$$

where $U_i = U(x)$ is the ambient wind velocity of the $i$-th particle and $v_i^\infty$ is the terminal velocity, which in general is a function of the attributes $a_i$ and the state of the ambient air.

The motion of a super-droplet is the same as that of a droplet, which is described in equation (2), and $v_t(t)$ is equal to the terminal velocity.

### 2.3 Condensation and evaporation

The condensation/evaporation process is based on Köhler's theory, which takes into account the solution and curvature effects on the droplet's equilibrium vapour pressure (Köhler et al., 1936; Pruppacher and Klett, 1997, chapter 13; Rogers and Yau 1989). The growth equation of radius $R_i$ is derived as follows:

$$R_i \frac{dR_i}{dt} = \frac{(S-1) - \frac{a}{R_i} + \frac{b}{R_i^3}}{F_k + F_d} \tag{3}$$

$$F_k = \left(\frac{L}{R_v T} - 1\right)\frac{L\rho_{liq}}{KT}, F_d = \frac{\rho_{liq} R_v T}{D e_s(T)} \tag{4}$$

where $S$ is the ambient saturation ratio; $F_k$ represents the thermodynamic term associated with the heat conditions; $F_d$ represents the term associated with vapour diffusion; the term $a/R_i$ represents the curvature effect, which expresses the increase in the saturation ratio over a droplet compared with that of a plane surface; the term $b/R_i^3$ represents the reduction in the vapour pressure due to the presence of

a dissolved substance, where $b$ depends on the mass of solute $M_i$ dissolved in the droplet; and $a \simeq 3.3 \times 10^{-5} cm \ K/T$ and $b \simeq 4.3 cm^3 i M_i/m_s$, where $T$ is the temperature, $i \simeq 2$ is the degree of ionic dissociation and $m_s$ is the molecular weight of the solute. $R_v$ is the individual gas constant for water vapour, $K$ is the coefficient of thermal conductivity of air, $D$ is the molecular diffusion coefficient, $L$ is the latent heat of vaporization, $e_s(T)$ is the saturation vapour pressure, and $\rho_{liq} = 1.0 \ g \ cm^{-3}$ is the density of

liquid water.

### 2.4 Collision-coalescence and the electric effect

In warm clouds, the collision-coalescence of two droplets to form a larger droplet is responsible for precipitation and the cloud lifetime. The droplet growth due to the coalescence is controlled by the net

action of various forces impacting the relative motion of the two droplets. The effective collision-



coalescence of droplets can be evaluated by the collision-coalescence kernel $K$, which can be described as follows:

$$K = E\pi\big((R_p + r_p)^2 |v_R^\infty - v_r^\infty|\big) + K_B \tag{5}$$

where $E = E_0(R_p, r_p) + E_{es}(R_p, Q_R, r_p, q_r)$ is the collision-coalescence efficiency, and $K_B$ is the Brownian coagulation kernel. $R_p$ represents the radius of the large droplet, $r_p$ is the radius of small droplets. Similarly, $Q_R$ represents the mean charge of large droplets, $q_r$ is the mean charge of small droplets. And $v_R$ represents the terminal velocity of large droplets, $v_r$ is the terminal velocity of small droplets.

$E_0(R_p, r_p)$ takes into account the effect of a small droplet/particle being swept by the stream flow around a larger droplet or bouncing on the surface by front, side or rear collection, or droplets of similar size collide on the downstream side and are caught (Davis, 1972; Hall, 1980; Jonas, 1972; Pruppacher and Klett, 1997, chapter 14). Following Seeβlberg et al. (1996) and Bott (1998), the collision efficiency of Davis (1972) and Jones (1972) for small droplets and the collision efficiency of Hall (1980) for large droplets are adopted. We assume the coalescence efficiency is unity in this study.

The Brownian coagulation kernel $K_B$ is given by Seinfeld and Pandis (2006, chapter 13) using Fuchs (1964) corrected factor to correct the boundary condition of absorbing particles surface. The Fuchs Form of Brownian coagulation coefficient is derived as follows:

$$K_B = 2\pi(D_1 + D_2)(D_{p1} + D_{p2})\left(\frac{D_{p1}+D_{p2}}{D_{p1}+D_{p2}+2(g_1^2+g_2^2)^{1/2}} + \frac{8(D_1+D_2)}{(\bar{c}_1^2+\bar{c}_2^2)^{1/2}(D_{p1}+D_{p2})}\right)^{-1} \tag{6}$$

Where:

$$\bar{c}_i = \left(\frac{8kT}{\pi m_i}\right)^{1/2} \tag{7}$$

$$\ell_i = \frac{8D_i}{\pi \bar{c}_i} \tag{8}$$

$$g_i = \frac{\sqrt{2}}{3D_{pi}l_i}\left[\big(D_{pi} + l_i\big)^3 - \big(D_{pi}^2 + l_i^2\big)^{\frac{3}{2}}\right] - D_{pi} \tag{9}$$

$$D_i = \frac{kTC_c}{3\pi\mu D_{pi}} \tag{10}$$

$l_i$ represents particle mean free path, $D_i$ represents Brownian diffusivity; $D_{pi}$ represents diameters of particles, $m_i$ is particle mass, $k = 1.381 \times 10^{-23} J/K$ is the Boltzmann constant, $\mu$ represents the viscosity rate of air and $C_c$ is a slip correction factor.

In this study, we propose a parameterization of the collision efficiency due to the electric force $E_{es}(R_p, Q_R, r_p, q_r)$ based on the work by Zhou et al. (2009) and Tinsley and Zhou (2015). The induced charge on the droplet is involved in our $E_{es}$. Based on the trajectory model simulation, the electric force with the IM treatment (Tinsley et al., 2006) and the CS treatment (Zhou et al., 2009) can significantly contribute to the collision efficiency. For droplets with opposite sign charges, in the front and side collision ranges, the short-range attractive electric force due to the induced image charge provides additional force to balance the repulsive force. In the rear collision range, this short-range attractive force contributes to balancing the inertia. The rear collision range is relevant for droplets smaller than 0.1 microns, where in our droplet charge setting by (16), the droplet accepts less than 1 element charge and the electric force does not have a significant effect on the collision process. Therefore, the main electric



force remains in the side and front collision range and droplets accept more than 1 element charge in this study.

The analytical parameterization for the collision efficiency with the electric force suggested by Davenport and Peters (1978) is used with modification to include the image charge effect of opposite charged droplets in our study. Tinsley and Zhou (2015) developed same charged droplets charge effect.

$$E_{es} = \frac{4c_f}{6\pi\mu_a R_p V(D_p)} \cdot F_{es} \tag{11}$$

where $c_f$ is the Cunningham correction factor, $\mu_a$ is the air viscosity, and $V(D_p)$ is the terminal velocity of the droplet. $F_{es}$ is the electric force between the collision droplets.

In this study, Fes is calculated in four different ways, namely, CB, IM, Khain04, and CS, which are given by eqs. (13-16), respectively. The distance parameter $r_{nt}$ is needed to calculate $F_{es}$. Based on the trajectory simulation results by Zhou et al. (2009), $r_{nt}$ is fitted as follows:

$$r_{nt} = \frac{r_p}{R_p}[1 + r_{ref} \cdot \left(\frac{R_p}{r_{p/2}}\right) + \frac{R_p}{r_p}] \tag{12}$$

where $r_{ref} = 0.01$, $R_p$ and $r_p$ represents the radius of large and small particles. $Q_R$ and $q_r$ are charge of large and small particles

CB treatment considers only the Coulomb force between the centre points of the droplets. Then, Fes is given by

$$F_{es} = \frac{1}{4\pi\varepsilon_0} \frac{Q_R q_r}{r_{nt}^2} \tag{13}$$

Where $\varepsilon_0 = 8.854 \times 10^{-12} Fm^{-1}$ is the dielectric permittivity of free space.

Khain04 used the superposition method to calculate a four-dimensional (with respect to droplet size and charge) lookup table for collision efficiency, and present an approximated solution for electrostatic forces of droplet by following formula:

$$F_{es} \approx \frac{Q_R q_r}{4\pi\varepsilon_0 r_{nt}^2} + \frac{1}{4\pi\varepsilon_0}\{Q_R^2 r_p \left[\frac{1}{r_{nt}^3} - \frac{r_{nt}}{(r_{nt}^2-r_p^2)^2}\right] + q_r^2 R_p \left[\frac{1}{r_{nt}^3} - \frac{r_{nt}}{(r_{nt}^2-R_p^2)^2}\right] + Q_R q_r R_p r_p \left[\frac{1}{r_{nt}^4} + \right.$$
$$\left. \frac{1}{(r_{nt}^2-R_p^2-r_p^2)^2} - \frac{1}{(r_{nt}^2-R_p^2)^2} - \frac{1}{(r_{nt}^2-r_p^2)^2}\right]\} \tag{14}$$

Note that in this study, we calculate collision-coalescence kernel of Khain04) method by eq (14) for electrostatic forces and eq (11) for charge effect, whereas Khain04 used 4-dimensional lookup table for collision efficiency.

When the large droplet radius is 100 times larger than the small droplet radius, the IM treatment, is accurate enough. $F_{es}$ for IM treatment is given by:

$$F_{es} = \frac{4Kq_r^2}{(d_p/2)^2}[\frac{r_{nt}}{(r_{nt}^2-1)^2} + \frac{1}{r_{nt}^2} \cdot (\frac{Q_R}{q_r} - \frac{1}{r_{nt}})] \tag{15}$$

where $K = 9 \times 10^9 (in\ Nm^2 C^{-2})$ .

If the ratio between the droplet and particle is less than 100, the electric force is treated by the CS method according to Zhou et al. (2009), which originates from Davis (1964a, 1964b):

$$F_{es} = 4K \cdot (q_r Q_R \frac{F_6}{r_p^2} + q_r^2 \frac{F_7}{r_p^2} + Q_R^2 \frac{F_5}{r_p^2}) \tag{16}$$

where $F_5$, $F_6$, and $F_7$ are dimensionless complex polynomial expressions given by Davis (1964a, 1964b) that depend only on the radii of the two droplets and their distance $r_{nt}$.



In this study, we assume that $J_z$ charges the droplets. Although the charging process of droplets of 10μm radius completes 70% in 680s (Zhou & Tinsley (2012)), in this study, we consider that the charge amount of the resultant droplets after collision-coalescence relaxes to their equilibrium value instantaneously.

Regarding the charge polarity, after convective mixing of inner cumulus clouds, the opposite charge droplets from the cloud boundary get into the cloud and keep opposite charge amount because of the relatively long timescale of discharge, with a significant impact on the early stage of rain droplets formation. The coalescence of large rain droplets is dominated by gravity settling. We will consider the extreme case that the charge polarity of the two droplets is always opposite. Note that Khain04 subtracted

the charge amount of opposite polarity particles after collision-coalescence, added the charge amount of same polarity particles respectively. In our study, following Andronache (2004), the mean charges on the large droplet and small droplet are described as a function of the droplet radii as follows:

$$Q_R = 4a\alpha D_p^2 \;\&\; q_r = 4a\alpha d_p^2 \tag{17}$$

Where $D_p$ and $d_p$ are the diameters of the large droplet and small droplet, respectively, $a =$

$\pi \times U_b \times \varepsilon_0 \times 10^{-2} = 0.83 \times 10^{-6}$ is two orders of magnitude smaller than the maximum particle charge, representing weakly charge condition, here $U_b$ is the air breakdown voltage and $\varepsilon_0$ is the dielectric permittivity of free space, and the charging rate $\alpha(cm^{-2})$ is an empirical parameter ($\alpha$ is referred to herein as the droplet charging rate, equal to ratio of particle charge amount and maximum possible charge) that varies between 0, which represents neutral particles, and 7, which represents highly electrified clouds

associated with thunderstorms (Andronache, 2004). In our work, the $\alpha$ value ranges from 0.1 to 0.6, which represents a weakly electrified cloud. Compared with the maximum charge of the droplet method used by Khain04, when $\alpha$ ranges from 0.1 to 0.6, the charge on the droplet reaches 0.3% up to 2% of the maximum charge, which is 10 times to 2 times smaller than the lowest value used by Khain04 and Wang et al. (2015). The minimum limit for the droplet charge is equivalent to 1 elemental charge. This could

be a reliable estimation for the accumulated charge on droplets with the downward current density ($J_z$) since a droplet with a radius of 10 μm can accept 200 element charges when $\alpha = 0.1$, which is consistent with the stratus cloud charge distribution simulation by Zhou and Tinsley (2007, 2012).

Figure 1 shows a comparison of the collision-coalescence kernel for droplets radii of 40 μm (black lines), 20 μm (green lines) and 10 μm (red lines) as a function of the small droplet radius with different

calculation methods. The solid line represents the calculation with the analytical method for the CS treatment, the dash-dot-dot line represents the calculation with the analytical method for IM, the dash-dot line represents the calculation with the analytical method for CB, , the long dash line represents the calculation with the analytical method for Khain04, the dotted line represents no static electric force and the dashed line represents the calculation with the trajectory method with CS. In Figure 1(a), the empirical

parameter (α) for the droplet charging rate is 0.2, and that in Figure 1(b) is 0.3. The results show that the main electro-coalescence range is approximately 0.1 μm to 10 μm. When the small droplet radius is less than 0.1 μm, the collision process is controlled by Brownian motion due to an excessively small number of charges on the small droplet. When the radius of the small droplet is larger than 10 μm, the collision process is controlled by gravity collision. The electric force has a larger effect on the smaller droplet.

The electric force treated with the CS method has a larger effect on the collision-coalescence kernel than that of the IM method and the CB method. In the range of the Greenfield gap, the collision-coalescence



kernels from the analytical method fit well with those with the trajectory method result. Note that CB, Khain04 and IM method does not take into account the collision of same size droplets, for CS method, the $Q^2$ term provide attractive or repulsive force between same size droplets ensures collision. For the range of droplets smaller than 10 μm, when the particle radius is close to 10 μm, CB, Khain04 and IM method deviates from the trajectory result, but the result of the CS method becomes over 2 times less than that of the trajectory method, where the collision process is controlled by the interception effect. Although the analytical method cannot reproduce the interception effect, it can give the lower limit of estimation to the effect of the electric force effect with the conducting sphere method.

**2.5 Fluid dynamics of moist air**

In our model, the warm cloud dynamics is described by the fully-compressible non-hydrostatic equation as follows:

$$\frac{\partial \rho}{\partial t} + \nabla \cdot (\rho U) = \frac{\partial \rho}{\partial t}|_{cm} \tag{18}$$

$$\frac{\partial \rho q_v}{\partial t} + \nabla \cdot (\rho q_v U) = \frac{\partial \rho q_v}{\partial t}|_{cm} + D_v \nabla^2 (\rho q_v) \tag{19}$$

$$\frac{\partial \rho U}{\partial t} + \nabla \cdot (\rho U \otimes U) = -\nabla P - \rho g \hat{z} + \frac{\partial \rho U}{\partial t}|_{cm} + \mu \nabla^2 U \tag{20}$$

$$\frac{\partial \rho \theta}{\partial t} + \nabla \cdot (\rho \theta U) = \frac{\partial \rho \theta}{\partial t}|_{cm} + \frac{k}{c_p} \nabla^2 \theta \tag{21}$$

$$P = \rho R T = P_0 \left(\frac{\rho \theta R}{P_0}\right)^{c_p/(C_p - R)} \tag{22}$$

Here, $\boldsymbol{U}=(U,V,W)$ represent wind velocity, $\rho_d$ is density of dry air, $\rho_v$ is density of water vapor, the density of moist air $\rho \coloneqq \rho_d + \rho_v$, $q_v \coloneqq \rho_v/\rho$ is specific humidity, $q_d \coloneqq \rho_d/\rho$ is mass of dry air per unit mass of moist air, T is temperature and P is pressure, $\theta \coloneqq T/\prod \coloneqq T/(P/P_0)^{R/c_p}$ is potential temperature of moist air, where $P_0 = 1000 hPa$ is the reference pressure; $R \coloneqq q_d R_d + q_v R_v$, $R_d$, $R_v$ are the gas constants of moist air, dry air and water vapor; ditto, $c_p \coloneqq q_d c_{pd} + q_v c_{pv}$, $c_{pd}$, $c_{pv}$ are the isobaric specific heats of moist air, dry air and water vapor. The model employed variable $G \coloneqq \{U, \rho, q_v, \theta, P, T\}$ to represent the state of moist air.

The four terms with the form $\partial \cdot/\partial t|_{cm}$ indicated cloud microphysics coupling terms. $\partial \rho/\partial t|_{cm} = \partial \rho q_v/\partial t|_{cm}$ represents source of vapor:

$$\frac{\partial \rho}{\partial t}|_{cm} = \frac{\partial \rho q_v}{\partial t}|_{cm} = s_v \tag{23}$$

where $s_v$ indicates the source of vapor through condensation/evaporation. $s_v$ can be evaluated by the microphysics variables as follows:

$$s_v(x,t) = -\sum_{i \in I_r(t)} \delta^3(x - x_i(t)) \frac{dm_i}{dt}\Big|_{cnd/evp} \tag{24}$$

Here $\delta^3(x)$ represent the three-dimensional Dirac delta function, and the time derivatives for condensation/evaporation.

$$\frac{\partial \rho \theta}{\partial t}|_{cm} = -\frac{L_v s_v}{c_p \prod} \tag{25}$$



$\partial \rho U/\partial t|_{cm}$ represents the drag force from the particles, as mentioned in 2.3: $F_i^{drg} = m_i g\hat{z} + d(m_i v_i)/dt$. The value of second term of $F_i^{drg}$ is much smaller then first term:

$$\frac{\partial \rho \mathbf{U}}{\partial t}\Big|_{cm} = -\sum_{i \in I_{r(t)}} \delta^3(x - x_i(t))\boldsymbol{F}_i^{drg}$$
$$\approx -\Big[\sum_{i \in I_{r(t)}} \delta^3(x - x_i(t))m_i(t)\Big]g\hat{z} \qquad (26)$$


2.6 Design of our numerical experiment

To evaluate the effect of electro-coalescence on warm clouds, 2D simulation of an isolated cumulus is performed following the setup of Lasher-Trapp et al. (2005). Note the original study of Lasher-Trapp et al. (2005) was conducted in 3D, but 2D simulation is used in this study to save computational resources.

The initial profile of the atmosphere is horizontally uniform. The vertical profile of the moist air is given by 1545 UTC 22 July sounding data from the Small Cumulus Microphysics Study (SCMS) in Florida. The cloud base is steady at 1050 m, and the maximum cloud top height is 5350 m. As suggested by Lasher-Trapp et al. (2005), wind shear is assumed to be absent, and random velocity perturbation is applied (maximum of 0.5 ms$^{-1}$) in the lowest kilometre of the model.

In general, there are different types of soluble/insoluble aerosols in a droplet. In the model, only one soluble substance ($(NH_4)HSO_4$ aerosol) is applied for simplicity. Initially, the aerosols are uniformly distributed in the simulation domain. The number and size distribution     are made by increasing the number concentration 3 times from that given in van Zanten et al. (2011) for RICO intercomparison case. The aerosol particles are composed of ammonium bisulfate, and the number-size distribution is given by

a bimodal log-normal distribution: The particle number concentrations of the two modes are $N1 = 3x90$ $cm^{-3}$ and $N2 = 3x15\ cm^{-3}$, respectively; the geometric mean radii are $r1 = 0.03\ \mu m$ and $r2 = 0.14\ \mu m$, with geometric standard deviations of $\sigma 1 = 1.28$ and $\sigma 2 = 1.75$, respectively.

**2.7 Numerical setup and schemes**

Shima et al. (2009, 2020) constructed a particle-based cloud model SCALE-SDM by implementing the SDM into SCALE, which is a library of weather and climate models of the Earth and other planets (Nishizawa et al., 2015; Sato et al., 2015; https://scale.riken.jp). Because of its efficient Monte Carlo algorithm for coalescence, the particle-base scheme SDM requires less computational cost to accurately simulate clouds and precipitation. We implemented the electro-coalescence process into SDM's

coalescence scheme as defined by Eqs. (5) -(17). The moist air fluid dynamics are calculated by Eqs. (18)-(26) using SCALE's dynamical core using a forward temporal integration scheme with an Arakawa-C staggered grid (Arakawa and Lamb, 1977) using a finite volume method.

For the initialization of the super-particle, the "uniform sampling method" is applied as in previous works (Arabas and Shima, 2013; Shima et al., 2014, 2020; Sato et al., 2017, 2018). Unterstrasser et al. (2017)

found that the uniform sampling method is more efficient than the "constant multiplicity method". Then, the multiplicity of the super-droplets becomes proportional to the initial distribution function of real particles:

$$\xi(a,x) = n(a,x,t=0)/(N_s(0)p), p(a,x) = p = constant \qquad (27)$$



In SCALE-SDM, moist air dynamics and cloud microphysics processes for aerosol/cloud/precipitation particles are integrated separately by using the 1$^{st}$-order operator splitting scheme. $\Delta t$ is set as the common time step. We set $\Delta t_{adv}, \Delta t_{cnd/evp}, and\ \Delta t_{coal}$ as the time steps for the advection and sedimentation of particles, condensation/evaporation, and collision-coalescence. We set $\Delta t_{dyn}$ as the time step for the fluid dynamics of moist air, which has to fulfil the Courant-Friedrichs-Lewy (CFL) condition of acoustic waves. All these time steps are divisors of the common time step $\Delta t$ . The order of calculation in the model is as follows: 1) calculate the fluid dynamics without the coupling terms from the particles to moist air, and update the moist air; 2) update the super-droplets $\{\{\xi_i, x_i, a_i\}\}$ from t to $t + \Delta t$. We integrate one cloud microphysics process one time step forward and then moves on to the next process. Process lags in time is calculated preferentially. Simultaneously, the feedback from particles to moist air comes through the coupling terms of (23-25), and we update the moist air from $G_{lmn}(t)\ to\ G_{lmn}(t + \Delta t)$.

In our simulation, the domain of the simulation is two dimensional (x-z), 10 km in the horizontal and vertical directions with 50 m grid spacing, and the calculation time steps are $\Delta t = 0.4$ s, $\Delta t_{dyn} = 0.05s$, $\Delta t_{adv} = 0.4s$, $\Delta t_{cnd/eva} = 0.1s$, and $\Delta t_{coal} = 0.2s$. Initial super droplet number concentration per grid cell is 128. Sub-grid scale turbulence model is not used in this study. To evaluate the fluctuation effect, a 50-member ensemble of simulations is conducted by changing the pseudo random number sequence.

## 3. Results

### 3.1 The effect of charged droplets on cloud evolution

Figure 2 shows a snapshot of the spatial structure of clouds at simulation time of 1500 seconds(s) 2100s and 2700s, which demonstrates the temporal evolution of the mixing ratio of cloud water and rainwater. Figure 2(a)-(c) shows the results with the no charge (NC) effect setting, Figure 2(d)-(f) shows the results of the charge effect depending on CB setting, and Figure 2(g)-(i) shows the results of the electric effect with CS setting, where the empirical parameter for the charging rate ($\alpha$) on the droplet is 0.3 and the minimum charging amount is 1 element.

The results show that with electro-coalescence by the CS setting (Figure 2(d)-(i)), the cumulus takes a shorter time to form rain droplets than with that by the CB and NC settings. Comparing Figure 2(d)-(f) and Figure 2(g)-(i) shows that the electric force with the CS setting has a much stronger impact on the cloud evolution than that with the CB setting. For the CS setting, there is heavy precipitation at 2700s, while there is only haze for the CB and NC settings.

Figure 3(a) shows the time evolution of the domain-averaged accumulated precipitation amount, where the solid line represents the NC setting, the dashed line represents the CB setting, the dotted line represents the IM setting and the dashed-dotted line represents the CS setting. The shade represents the standard deviation error, which is calculated from the 50 members of the ensemble of CS, CB and NC treatments. Figure 3(b)-(d) shows the domain-averaged pathway, including the total liquid water pathway (b), rainwater pathway (c), and cloud water pathway (d); in the figures, the solid line represents NC



setting, the dashed line represents CB setting, the dotted line represents the IM settingand the dashed-dotted line represents CS setting. The grey shades indicate the standard deviation error, which is also calculated from the 50 members of the ensemble. An unbiased estimator is used to calculate the standard

deviation error. The results show that the accumulated precipitation amount in the CS setting is 52.5% higher than that in NC, 34.9% higher than CB settings and 8.4% higher than IM settings. There is significant difference between the accumulated precipitation amounts in NC setting, CS setting, IM setting and CB setting. The initial precipitation time for four settings are all start at 2100s. However, the total liquid water path and cloud path of the CB and NC settings are significantly higher than those of

the CS setting because higher precipitation eliminates cloud evolution.

Figure 4 shows the evolution of the mass density distribution of droplets for NC setting (solid line), CB setting (dashed line), IM setting (dotted line) and CS setting (dashed-dotted line) at 1500s, 2100s and 2700s. At these three stages, the droplet size distribution in the CS setting is much wider and rain droplets are much coarser than in the NC, IM and CB settings. At 2100s, there are two mass density peaks of 10

μm droplets and 1000 μm droplets for the NC, IM, CS and CB settings, while for the CS setting have highest mass density of 1000 μm, which is consistent with the results of Figure 3.

### 3.2 The effect of charge on droplets

Figure 5 shows the results of droplet evolution (a)-(c) and the water fraction path (d)-(f) for charging rate ($\alpha$) is 0.1 (solid line), 0.2 (dashed line), and 0.6 (dashed-dotted line); the grey line represents the NC

setting. The results show that there are no significant differences between the results of the NC and CS settings with the charging rate $\alpha=0.1$, which gives 0.3% of the maximum charge on the droplets. With the enhancement of the charge on the droplets, clouds can form more rapidly. When the charging rate $\alpha=0.6$, at 1500s, there are larger droplets with radius over 1000 μm and even droplets sized 5000 μm. However, because a higher charging rate causes faster cloud elimination, at 2700s, a lower charging rate

condition results in a higher droplet mass density at a peak of approximately 1000 μm, which indicates that a higher charging rate results in a shorter lifetime of the cumulus cloud. The results of the domain-averaged water path in Figure 8 (d) - (f) are consistent with those in Figures 8 (a) - (c).

Figure 6 shows the domain-averaged precipitation amount as a function of the droplets charging rate in the CS setting. Similar to the results in Figure 5, droplets with higher charging rate produce precipitation

earlier than those under low charging rate conditions. With the enhancement of the charging rate, the precipitation amount at 3500s does not simultaneously increase under all conditions. When the charging rate is 0.6, the final precipitation amount decreases due to more liquid and cloud water loss in the early stage of cloud formation. In Figure 6, the result of the CS setting charging rate $\alpha$ equal to 0.05, which is 0.16% of the maximum charge on the droplets, is given by the solid orange line, which the precipitation

amount is 9.5% higher than that of the NC setting.

### 3.3 The effect of the aerosol concentration

Figure 7 shows the results for the CS setting of the domain-averaged precipitation amount as a function of aerosol concentrations of $3.15 \times 10^8/m^3$ (solid line), $9.45 \times 10^8/m^3$ (dotted line) and $15.75 \times 10^8/m^3$ (dashed-dotted line), which represent low aerosol (LA), medium aerosol (MA) and heavy aerosol (HA)



concentrations, and the charging rate ($\alpha$) is 0.2. The grey lines represent the results of LA, MA and HA
conditions with the NC setting. For the LA and MA conditions, with increasing aerosol concentration,
the appearance of precipitation occurs earlier, and the precipitation amount increases due to more cloud
nuclei. Under the HA conditions, the enhancement of the aerosol concentration causes low precipitation
due to the low effective droplet radius. Compared with the enhancement rates due to the electrostatic

force, the higher the aerosol concentration is, the greater the enhancement rate of the domain-averaged
precipitation. Under HA conditions, the enhancement rate of precipitation is 782.4% higher than that
under the NC setting; the enhancement rate under MA is 467.2% higher, and that under LA is 110.6%
higher.

### 3.4 Comparison of different electrostatic force calculations

Figure 8 shows the results of the domain-averaged precipitation amount under different electrostatic force
settings, and the charging rate is 0.3. The grey solid line represents NC setting, and the dashed line
represents CB setting. The dotted line represents droplets set to opposite sign charges with electrostatic
force by the IM charging method (IM), which means that the induced charge appears only on large
droplets due to the charge accumulated when small droplets collide. The black dashed-dotted line

represents droplets of set to opposite charges with electrostatic force by the CS method (CS). The yellow
dashed line represents the result of droplets with opposite sign charges and the setting based on Khain04
method. The upper limit of the charge on the small droplets is 50 element charges, the lower limit of the
limit of the charge on the droplets is 1 element charges and the orange dashed-dotted line represents the
special setting of only a large droplet charged with electrostatic force by the CS method (CS-q0). For the

CS, Khain04 and IM setting, precipitation increases at 2100 s, which is 300 s before the NC setting. The
domain-averaged precipitation amount with the CS setting is 52.5% higher than that with the NC setting;
with the Khain04 setting, it is 5.42% larger; with the CS-q0 setting, it is 9.6% larger; and with the IM
setting, it is 8.45% larger.

### 4. Discussion

When the two droplets move together and coalesce, there are three sites where collisions can occur, the
front, side and rear. The radius ratio between a large droplet and a small droplet (RARA) controls the
collision site, and when the radius of the small droplet is less than 0.1 µm and the RARA is larger than
100, the collision is a rear collision. For front and side collisions, in clouds where the droplet size is less
than 40 µm and the relative humidity is 100%, the droplet collision is controlled by the balance of the

Stokes drag of the air flow and electric force. The analytic expression in our work suggested by
Davenport and Peters (1978) can give a good estimation, especially for the Greenfield gap part, although
in the front and side collision regions when the RARA is close to 1, the additional contribution due to
the interception associated with the electric effect cannot be fully reproduced by this method. When the
radius of the small droplet is less than 0.1 µm and the RARA is larger than 100, the collision process is

controlled by the rear collision. For the rear collision, the flow drag, electric force and Brownian motion
of the small droplet can impact the collision process. In the present work, because the charge on the





droplet varies as a function of the droplet radius, there is less than 1 element charge on a droplet with a radius less than 0.1 μm. Therefore, the electric force does not have a significant effect on the collision process even for droplets of opposite signs, and the Brownian collision efficiency could be good enough

for estimation under these conditions. When the amount of charge on a small droplet is over several element charges due to the evaporation of a large droplet with a large amount of charge, the rear collision could be significantly affected by the electric force (Tinsley and Leddon, 2013). The net attractive force of droplets with opposite signs increases the collision efficiency, and the net repulsive force of droplets with the same sign decreases the collision efficiency; this is called electro-anti-coalescence. As Tinsley

(2001) mentioned, below the cloud bottom boundary, there could be a highly charged nucleus or small droplet with tens to hundreds of element charges due to the evaporation of a highly charged droplet. These highly charged small droplets or nuclei could be moved into the cloud by upward air flow, which is not considered in this paper and will be evaluated in our next paper.

    In clouds, there are several ways to charge droplets, and in the cloud boundary, due to charging by the

vertical electric current density ($J_z$) from the ionosphere to the ground surface, a droplet in the cloud top boundary accumulates a positive charge, and that in the cloud bottom boundary accumulates a net negative charge; this has been shown by simulations (Zhou and Tinsley, 2007, 2012) and field observations (Nicoll et al., 2016). With the charging rate of 0.05, there are 103 element charges on a droplet with a radius of 10 μm, which is consistent with observation (Beard et al., 2004) and simulation

(Zhou and Tinsley, 2007) results. Therefore, in the stratus cloud, most droplet collisions occur between droplets of the same sign or between one charged droplet and one uncharged droplet. Using the CS method, the additional electrostatic force due to the multiple image dipoles between the colliding droplets can be addressed, even if the droplets have the same sign charges or a small droplet is uncharged. Khain et al. (2004) evaluated the electro-coalescence effect on warm clouds rain enhancement and fog

formation based on the image charge method from one induced dipole on each droplet. Zhou et al. (2009) claimed that when the RARA is close to 1, the collision efficiency calculated by the CS method, which treats the multiple induced dipoles on each droplet, is two factors larger than those calculated with the IM method. Therefore, for the Greenfield gap region and interception region, the evaluation of the charge effect with the CS method is more accurate. In Figure 8, due to the additional induced image charge on

droplets, the maximum averaged precipitation amount of the CS setting is 8.45% larger than that of the IM setting. It could be suggested that the CS method for electrostatic force should still be involved in the cloud model, although its computation time is 30% longer. The previous work by Khain et al. (2004) evaluated electro-coalescence at the lowest charging rate of 5% of the maximum charge on droplets. In our simulation, a charging rate ($\alpha$) of 0.05 up to 0.6 is tested, which represents 0.15% to 1.8% of the

maximum charge on the droplet. With a charging rate ($\alpha$) equal to 0.05, the electric force evaluated by the CS setting can increase the domain-averaged precipitation by approximately 9.5% compared to that of the NC setting, which provides the lower limit of the effect of electrostatic force on cloud formation. Tinsley et al. (2001, 2006) and Zhou et al. (2009) claimed that the induced charge on droplets of the same sign could produce a short-range attractive electrostatic force that increases the collision efficiencies.

The charge on the large droplets could exert an additional short range attraction on the small droplet, even if there is no charge or the same charge on the small droplets. However, for droplets of the same





sign, the short-range electrostatic force has a significant effect only if the charge ratio between the large droplet and small droplet $Q{:}q$ is greater than 100 or $q{:}Q$ is greater than 1. For $Q{:}q$ ratios larger than 100, the additional image charge effect on the small droplet due to the large charged droplet controls the

collision process. For $q{:}Q$ greater than 1, the additional image charge effect is due to the small charged droplet.

An increase in the aerosol concentration decreases the effective radius by increasing the concentration of small droplets, which could have a significant impact on cloud formation (Rosenfeld et al., 2008). According to our simulation, the effect of electro-coalescence is sensitive to the aerosol concentration.

With a high aerosol concentration, the average precipitation with an electric effect could be a factor of 4 higher than that of the NC condition. A much higher aerosol concentration corresponds to a more sensitive cloud response to the electrostatic force. Then, under high aerosol concentration conditions, a small variation in $J_z$ could have a significant effect on cloud formation. Alternatively, in highly polluted fogs or clouds, placing a small number of charged aerosols or droplets accelerates fog elimination or rain

enhancement due to electro-coalescence.

**5 Conclusion**

The electro-coalescence effect on a weakly electrified warm cumulus was revisited. Assuming opposite sign droplets charge by $J_z$ instantaneously, the charge amount determined by size of droplets. A new

simulation with the exact treatment of the electrostatic force for opposite sign charge case provides a good estimation of the effect of electro-coalescence in the Greenfield gap region. In the simulation, droplets smaller than 0.1 µm are controlled by Brownian motion. The results show that for droplets of opposite signs with the same treatment of the electrostatic force, the cloud evolution can be significantly changed as a function of the charging rate ($\alpha$). The same sign charge droplets case (Tinsley and Zhou

(2015)) and charge amount prediction are necessary for accurate simulation, we leave them for the future work. Electro-coalescence has a larger impact on highly polluted warm clouds or fog. This indicates that the effect of the electrostatic force with exact treatment should be included in cloud, weather and even climate models to improve the simulation accuracy.

**Code and data availability.** The source code of SCALE-SDM 0.2.5-2.3.0 and single simulation data of α=0.3 in four settings of NC, CB, IM, and CS are available from https://zenodo.org/records/10400635 (Zhang,2023). All the data used for this study can be reproduced by following the instructions included in the above repository. The data are also deposited in local storage at the University of Hyogo, Japan, and are available from the corresponding author upon request.


**Author contributions.** All the authors designed the model and numerical experiments. LM and SS developed the model code and RZ performed the simulations. RZ prepared the manuscript with contributions from all co-authors.



**Competing interests.** The authors declare that they have no conflicts of interest.

**Financial support.** This work was funded in part by the Strategic Priority Research Program of CAS (Grant No. XDB 41000000), the National Science Foundation of China (41971020), MEXT KAKENHI (grant no. 18H04448), JSPS KAKENHI (grant nos. 26286089, 20H00225, 23H00149), JST [Moonshot
R & D][Grant Number JPMJMS2286], China Scholarship Council (202106140113).



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



Figures and Captions:

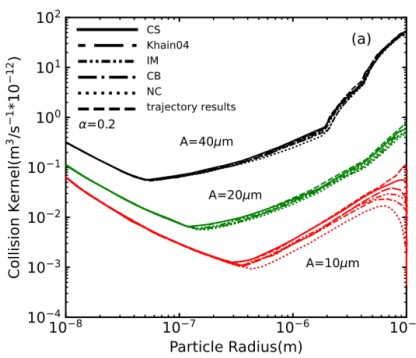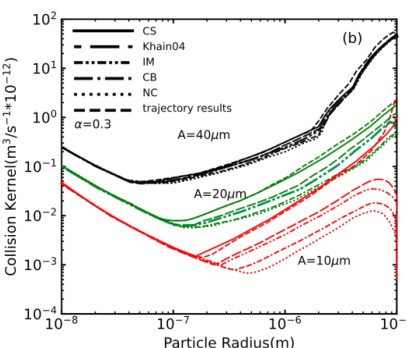

**Figure 1: Comparison of the effect of electric charge on the collision kernel for droplets sized 40 μm, 20 μm**
**and 10 μm with small droplet radii between 10⁻² μm and 10 μm. The charging rate $\alpha$ is 0.2 for panel (a) and**
**0.3 for panel (b). The solid line represents the results where the collision kernel is calculated by the analytical**
**expression and treats the charged droplets as CS setting. The long-dashed line represents the results**
**calculated by the analytical expression and treats the charge droplets as Khain04 setting. The dashed-dotted-**
**dotted line represents the results calculated by the analytical expression and treats the electrostatic electric**
**force by the IM setting. The dashed-dotted line represents the results calculated by the analytical expression**
**and treats the electrostatic electric force by the CB setting. The dotted line shows the results with NC setting.**
**The dashed line represents the results of the trajectory simulation according to Zhou et al. (2009).**



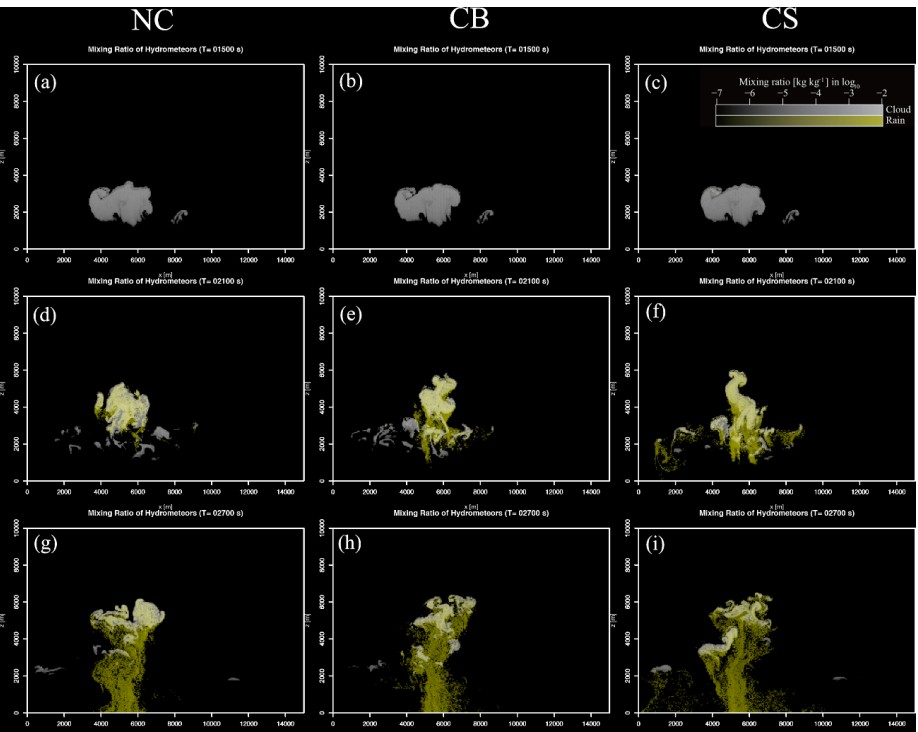


**Figure 2: A comparison of the spatial structure of the mixing ratio of hydrometeors of the cumulus with NC setting, the electric force evaluated by CB setting and the electrostatic electric force evaluated by CS setting at times of 1500s, 2100s and 2700s. The charging rate α is 0.3.**






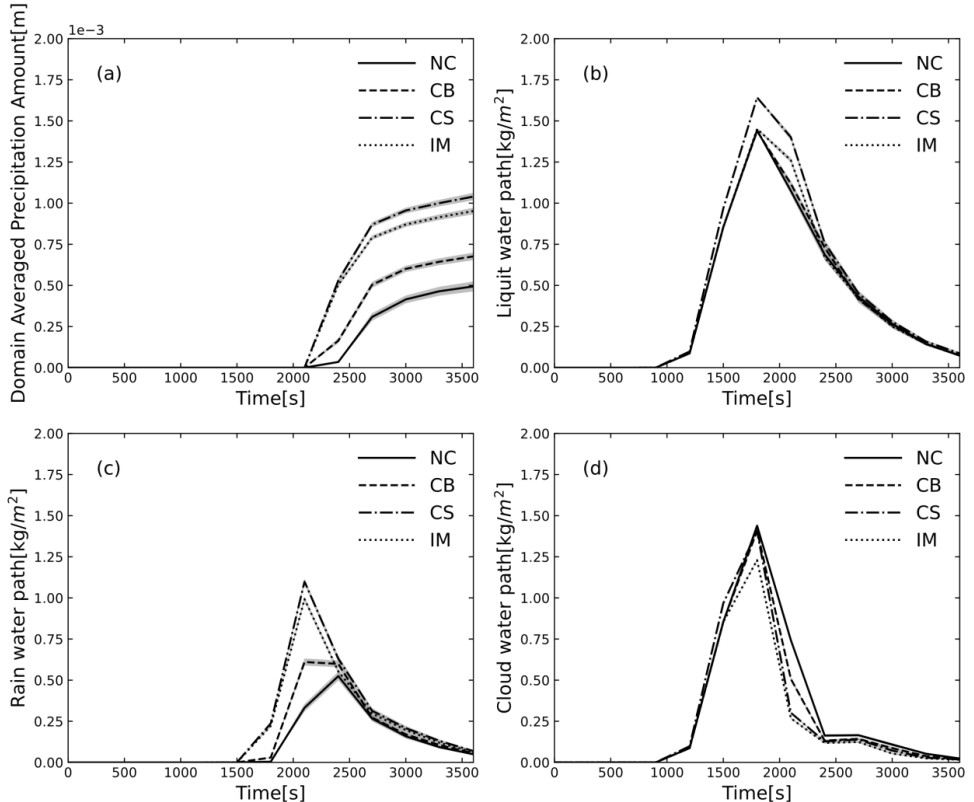

**Figure 3: The time evolution of the domain-averaged precipitation amount (a) and the domain-averaged water path of the liquid water path (b), rainwater path (c) and cloud water path (d), which is consistent with Figure 2. The solid line represents NC setting, the dashed line represents CB setting, and the dashed-dotted line represents CS setting. The grey shade indicates the standard deviation calculated from 50 members of the random ensemble.**








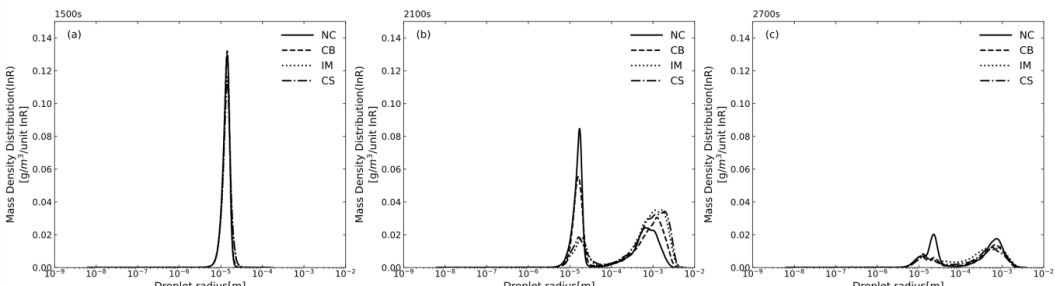

**Figure 4: The mass density distribution evolution of the droplets at 1500s (a), 2100s (b) and 2700s (c), which is consistent with Figure 2 and Figure 3. The solid line represents NC setting, the dashed line represents CB setting, the dotted line represents IM settings and the dashed-dotted line represents CS setting.**

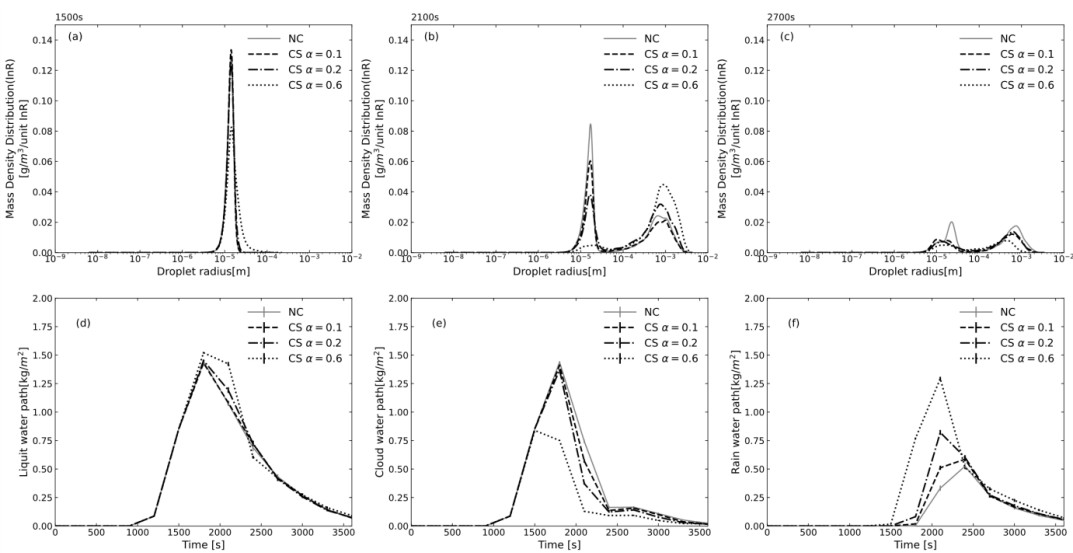

**Figure 5: Comparison of the cloud evolution for variable charging rates. The mass density distribution of droplets at 1500s (a), 2100s (b) and 2700s (c) and time evolution of the domain-averaged water path of liquid water precipitation (LWP) (d), cloud water precipitation (CWP) (e), and rain water precipitation (RWP) (f) are presented for charging rate α is 0.1 (solid line), 0.2 (dashed-dotted line) and 0.6 (dotted line).**



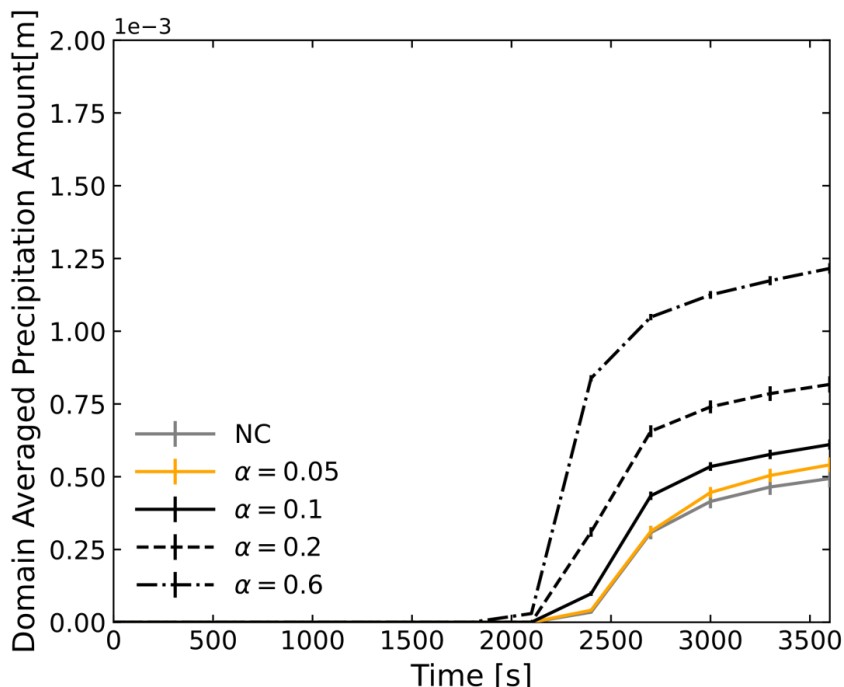


**Figure 6: A comparison of the evolution of the domain-averaged precipitation amount with variable droplet charging rate α is 0.05 (orange solid line), 0.1 (black solid line), 0.2 (black dashed line), and 0.6 (black dashed-dotted line), and the electric force is evaluated with the CS setting. The grey solid line represents the NC setting.**





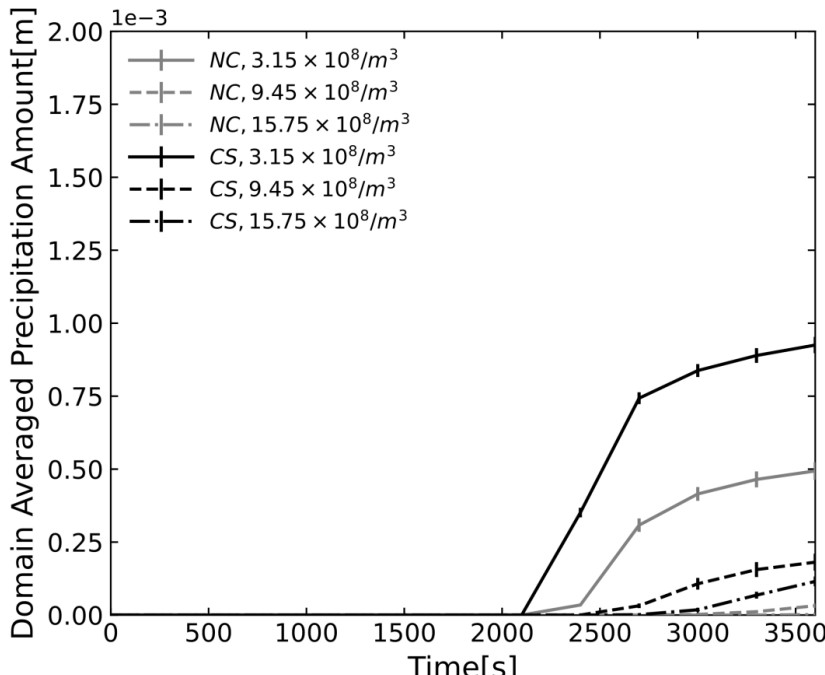

**Figure 7: The time evolution of the domain-averaged precipitation of the aerosol concentration represented by solid lines (3.15×10⁸/m³), dotted lines (9.45×10⁸/m³) and dashed-dotted lines (15.75×10⁸/m³). The black lines represent the simulation with the electric force, which is evaluated with the CS setting, and grey lines represent the simulation with NC setting. The charging rate α is 0.2.**





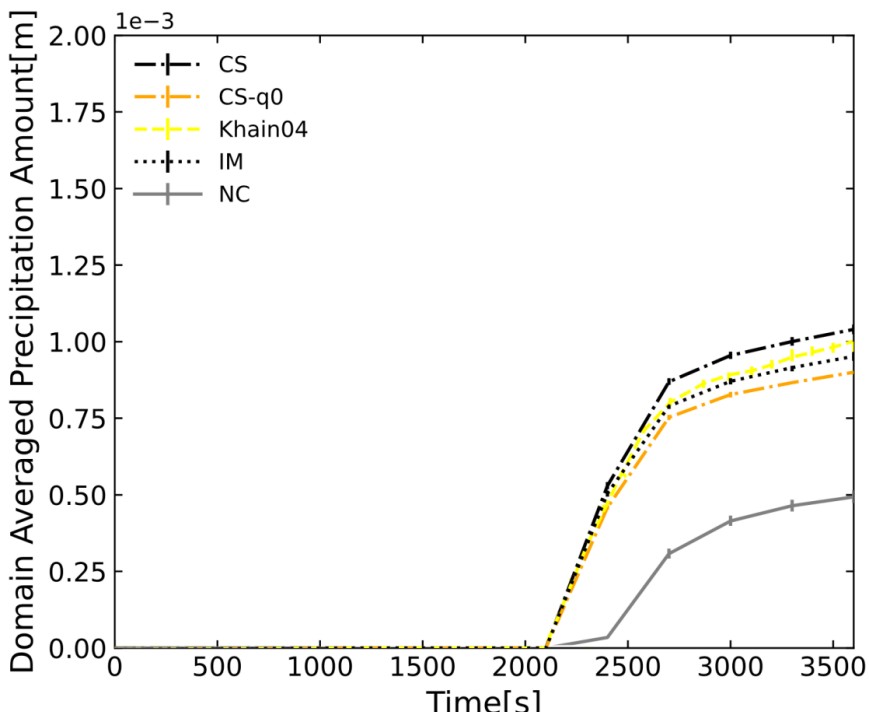

**Figure 8: Comparison of the time evolution of the domain-averaged precipitation amount for variable evaluation of the electric force. The grey solid line represents the NC setting. The dashed-dotted lines represent the results of the electric force evaluated by the CS method. The black dashed-dotted line represents the result of droplets of with opposite sign charges. The orange dashed-dotted line represents the condition where the charge is only on the large droplet, the yellow dashed line represents the result of droplets with**

**opposite sign charges and the CS method based on Khain et al. (2004). The black dotted line is the result of droplets with opposite sign charges, and the electric force is evaluated by the image charge method.**