# Peer review of "Preliminary evaluation of the effect of electro-coalescence with conducting sphere approximation on the formation of warm cumulus clouds using SCALE-SDM version 0.2.5-2.3.0"

_EGUsphere, 2023_

## Referee Comment (RC1)

**General Comments**

This study investigates the impact of enhanced coalescence due to droplet charge in high-fidelity simulations using the superdroplet method. While the methodology and results are thoroughly presented, the authors' key assumption of instantaneous droplet charging is not well-founded or discussed, and likely leads to a strong overestimate of the impact of droplet charge on precipitation. Similarly, the authors do not attempt to disentangle purely microphysical effects from flow field variability that stems from microphysics-flow coupling (vs. using a method like piggybacking). These reservations about the scientific merits combined with more minor concerns about the writing itself (including typos and inappropriate references) lead me to recommend major revisions before this paper be considered for publication.

**Specific comments**

1. The selected references for describing the importance of droplet coalescence (e.g. line 35) are not appropriate general references for this statement. For instance, Rosenfeld 2008 specifically concerns the a controversial mechanism of aerosol convective invigoration, which has relatively little to do with coalescence. Craig 1995 discusses radiative effects which are also not inherently specific to coalescence. Forbes & Clark does not even appear in the references. More appropriate citations would include review papers, chapters from the IPCC or a classical cloud microphysics textbook, or studies which specifically investigate droplet coalescence. Likewise in line 483: the stated impact of increasing aerosol concentration is the Twomey effect, and should not be attributed to Rosenfeld 2008!

2. I have some issues with your notation and definitions in section 2.4. Line 144: Given that both E0 and Ees are presumably efficiencies varying between 0 and 1, they should be multiplicative rather than additive. I am also confused by the notation of Rp, rp, QR, and qr, as the text describes these radii and charges as being general to "large droplets" or "small droplets", whereas I would imagine them to be descriptive of the larger droplet R and smaller droplet r for a given pair (R, r). What does the subscript "p" refer to?

3. The assumption in 212-213 that droplets charge instantaneously following coalescence seems like a major flaw in this study which would lead to a strong overestimate of the effects of electro-coalescence in an LES. Given that you are using the superdroplet method for this study, you should in fact be able to model the time response of charging on a given superdroplet as an additional attribute! This would provide a much more trustworthy study of the effects of droplet charge on coalescence that could actually be used to quantify and suggest whether this effect is notable. As it stands however, this assumption undermines the findings of this study and is not adequately discussed as a limitation or confounding factor in the abstract or conclusions. Furthermore, the values chosen for alpha in the numerical experiments are not well-justified with values or ranges measured in real clouds.

4. I also take issue with the comparisons made between different simulations given that the flow-field and microphysics are fully coupled. A more appropriate way to analyze the purely microphysical effects of electro-coalescence would be through the common technique of piggybacking, as other studies have shown that differences due to small perturbations to the flow field often outpace differences related to microphysics effects.

I do like the approach of using 50 ensemble members to analyze statistics of the superdroplet simulations, but I'm not convinced that this would help isolate microphysics from flow-field variability. Furthermore, in section 2.5 and 2.6, is it not clear whether the simulations performed are DNS or LES as there is no sub-grid scale turbulence model (line 340), nor is there any mention of what impact neglecting SGS turbulence would have on the results.

5. In general when discussing figures and results, details from the figure caption which describe the various lines are repeated in the full text unnecessarily (e.g. lines 240-245, line 356-364, and elsewhere throughout). This repetition should be removed and avoided.

6. Lines 401-406 suggest that the trend going from LA to MA conditions is opposite from the trend going from MA to HA conditions, when the figure in fact indicates that the trend is consistent. Increasing the aerosol concentration appears to uniformly delay and reduce the precipitation quantity.

**Technical comments**
- Line 33: "play a key role in cloud formation" should be "rain formation"
- Line 62: the Greenfield Gap should be concisely defined
- Line 200: there is an extra close-parentheses ")" after Khain04
- Line 203: there is an extraneous comma after "the IM treatment"
- Line 228: if alpha is a ratio, it should be unitless and have a maximum value of 1. Can you clarify the definition here?
- Line 305-306: Why are the number concentrations written this way, as "3 x" something?
- Line 312-313: "SDM requires less computational cost…" compared to what? Be specific; it requires more computational cost than a bulk method!
- Equation 27: what is p?
- Lines 433-435 are repeated from earlier in the paragraph.
- Lines 485-490 are a very good summary of key findings and insights from this study

---

## Referee Comment (RC2)

**Outline of the contents**

The reviewed paper focuses on numerical modelling of particle coagulation in a Cumulus cloud typical of fair-weather convection. The focus is on studying the effects of droplet charges on the effectiveness of coagulation, and the resultant changes in rainfall properties. The study employs a two-dimensional idealised fluid dynamics setup resolved on a grid with 50m spacing, with no subgrid-scale dynamics representation. The particulate phase is represented with simulation point particles, each representing a sample of modelled droplets. The simulation particles undergo collisions (only with other simulation particles, not within the subpopulation represented by each of them), and the resultant coagulation efficiency is parameterised taking into account theoretical considerations for the enhancement of collisions probability due to the particles being charged. The simulations do not involve tracking particle charges - these are a priori determined as a function of the droplet size. Particle collisions are treated employing coagulation kernel approach, without assessing the inter-particle distances. Besides collisions, the particles are subject to condensational growth and evaporation, sedimentation and advection with the flow. The representation of collisions and the initial sampling of the simulation particle population is probabilistic, and the study analyses 50-member ensembles for each setup.

**Overall impression and suggestion of a major revision**

Unfortunately, it is hard not to begin with pointing out that the lax approach to text and figure composition, reference consistency and equation typesetting distracts from the paper's content and tarnishes overall impression. The manuscript was submitted prematurely and calls for a major revision. There are (details below): unlisted and uncited references, symbol conflicts, omitted symbol subscripts, unreadable figure elements, and large parts of text that repeat from literature descriptions of model details that are of little relevance to the study (e.g., sections 2.3 and 2.5). There are several aspects which warrant elaboration or more substantial grounding (and literature references):

- The choice of the particle-resolved method is not explained - what are the benefits, tradeoffs, limitations as compared to other modelling techniques, in the very context of modelling charged-particle interactions?

- What are the implications of one of the key assumptions, namely that the drop charge is merely parameterised as a function if its size?

- Unlike the velocity-differential driven gravitational coagulation, the Brownian mode applies to same-sized particles, and thus should occur also within particles represented by a single simulation particle, which IIUC is not the case in the model?

- Despite the whole paper focuses on fair-weather convective cloud simulation, and only the very last sentence of the Discussion section relates to fog, the Conclusions section puts forward a hypothesis that "Electro-coalescence has a larger impact on highly polluted warm clouds or fog" - are there grounds for this statement in this study?

- More background on electro-coalescence would help to cater to readers not acquainted with the earlier works of the authors, and to highlight the importance of the results by explaining a broader perspective on the challenges in this domain, from measurement, theoretical and numerical modelling perspectives.

The paper clearly matches the journal scope. The paper is accompanied by open-source code and execution scripts fulfilling the journal requirements (although, there is certainly room for improvement - see comments below).

**Abstract**

- the first sentence of the abstract does not seem to apply to this paper - the analytic expression is not established here, it is used here? (if not, please clarify in the text)

- the abstract should clearly state what kind of simulations are done (2D, flow-coupled, capturing aerosol microphysics, warm rain, no subgrid dynamics, ...)

- it should be indicated that despite employment of a particle-resolved microphysics representation, the particle charges are not among the particle attributes

- similarly, worth clarifying that despite calculating drop trajectories, collisions are modelled assuming well-mixed coalescence volume assumption

- worth iterating the considered options of assumptions regarding charge treatment (e.g., charge polarity always opposite, ...)

- provision of numbers with 3 significant digits in the abstract (e.g., 5.42% higher) seems overzealous

- it is worth to highlight the probabilistic nature of the simulations and the number of realisations employed in the study

- usage of the word "treatment" in the abstract suggests that CS is an alternative to super-droplet method (line 17)

- according to the GMD guidelines (https://www.geoscientific-model-development.net/submission.html#manuscriptcomposition), references should not be included in the abstract unless urgently required - suggest removing the reference to Khain 2004 (retaining the rest of the sentence)

**Code and data availability**

- the Zenodo archive contains four 9GB tar files without any annotation or metadata, with two-letter file names – one can guess that these contain simulation output, but provision of a description that can be accessed prior to downloading it would be helpful;

- the referenced source code archive on Zenodo contains spurious non-portable compiler output (*.o, *.a, and *.mod files) which should be removed

- the title mentions v0.2.5-2.3.0 but the README.md file gives v0.0-2.2.2 - please make it consistent and add to Zenodo metadata

- the "The data ... are available from the corresponding author upon request" statement is not in line with the journal policies - the statement implies lack of anonymous and persistent access to the data and should be removed, while clarification on what is included in the multi-gigabyte datafiles on Zenodo provided

- Trying to understand the contents of the provided `contrib/SDM/sdm_coalescence.f90` file, I became puzzled with why all the electro-coalescence efficiency calculation lines (824, 825, 828, 830 and 834) are commented out, only to realise that the **README.md file hints that such code blocks need to be manually uncommented if trying to reproduce the simulation results** - it seems to be quite an obscure way to provide the code. Please provide the source code in a way that no manual alterations are needed to reproduce the results and that the version identifier provided matches the unmodified code used for simulations.

**Figures**

- Fig 1: the kernel dimensionality should be volume over time, not volume over inverse-time

- typo in Fig 3b and 5d: "Liquid" not "Liquit"

- font sizes in the figure labels (incl. axis tick labels) should match text size, and be uniform, currently these span all sizes from unreadable small (Fig. 2!) to unreasonably large

- the employment of a multiplication factor given above Y axis in figs 3a and 6-8 is misleading, non-standard and hard to notice, please use intuitive units instead

- labels are missing spaces between quantity name and unit parentheses, parentheses are sometimes "()", sometimes "[]"

- the choice of X and Y axis ranges in figs 3a, 6-8 seems awkward (vast majority of the plot area is left blank for no good reason)

- some units are typeset in italics, some in normal font

- the "*$10^{-12}$" multiplication factor for Y axis in Fig. 1 is awkward given the $10^{-4}$ - $10^2$ axis range - please avoid using scaling factors in labels

- X axis in figs 6-8 would be more intuitive if presented in minutes/hours, while Y axis in mm

- figure quality is poor due to choice of inadequate raster graphics format, please use vector graphics

**Equations**

- eq. 27 - split into two equations - the comma is hard to interpret

- parentheses in equations are typeset with misleading sizes (small parens surrounding large parens)

- using almost identical "a" and "$\alpha$" right next to each other is very misleading (e.g., eq. 17)

- suggest avoiding defining parameters with their units, e.g.: "$\alpha$ (cm-2)" and mentioning unitless values in the text – it is clearer to write "rate $\alpha$ ... with a range of $0.1 - 0.6$ cm$^{-2}$ ..."

**Text comments**

- line 26: "electrostatics on charged droplets" seems awkward - please rephrase

- line 37: cite Rayleigh 1879 (https://www.jstor.org/stable/113853)?

- line 62: suggest replacing well-known with so-called

- line 72: 0-dimension but 4-dimensional

- line 91: remove "accurate"?

- line 100: why the radius is "equivalent"? (+unintentional "::" at line end?)

- line 104: replace "floating in the atmosphere" with "in the domain"?

- line 109: "sediments" -> "sedimentation"

- line 114: likely worth referencing Montero-Martínez et al. 2009 (https://doi.org/10.1029/2008GL037111)

- line 126: "heat conditions" -> "latent heat release"

- line 131: mixing coefficients, units and functional dependencies in one expression leads to ambiguities - e.g., what does K means here - thermal conductivity (should be in italics) or Kelvin (all units should not be in italics, but with upright font)

- line 134: the water density was already defined in line 111

- line 143: parenthesis in eq. (5) suggest at first sight that $\pi$ is a function and E is a constant, while the opposite holds - please try to format the equations intuitively

- line 146: the single-particle size is given along with "mean" charge of particles - how the mean is defined, why isn't it a mean size as well? Should it be characteristic instead of mean? - please elaborate on the assumptions needed to define the notion of the kernel in four dimensional attribute space

- line 175: eq. (16) is referenced before being given

- line 183: different symbol used for terminal velocity than before

- line 184: collision -> colliding

- line 185: Fes -> $F_{es}$

- line 206: K was defined differently in line 133

- line 236: element -> elemental (also in lines 418 & 437)

- line 257: "the interception effect" was never mentioned before

- line 302: please rephrase "number and size distribution are made"

- line 304: the composition was already given in line 301

- lines 305-307: subscripts, units with upright font; also: worth mentioning that these parameters are actually altered across simulations (section 3.3)

- line 310: it would be worth to clarify that Shima et al. (2009) includes warm-rain algorithm definition, while the 2020 paper includes mixed-phase extension (not used here) and coupling with SCALE (used here)

- line 312: IIUC, the employed/developed SDM code is not available at the provided riken.jp URL - worth clarifying

- line 313: provide reference for the lower cost mention

- line 330: this reads as if the fluid dynamics were not influenced by the latent heat budget of the particles, but surely are - should there be a third step defined?

- line 333: I fail to understand the "Process lags in time is calculated preferentially" statement

- line 340: refer to Lu & Shaw 2015 (DNS study, https://doi.org/10.1063/1.4922645) and elaborate on the implications of lack of small-scale turbulence representation

- line 351: please underline that this is just one of 50 realisations simulated, and given that and the fact that these plots are hardly distinguishable, are the conclusions really supported by this figure?

- line 363: should be "standard deviation", not "standard deviation error", right?

- lines 365-366: are three significant digits for a percentage increase meaningful here?

- line 388: is it both domain- and ensemble-averaged? (same remark for lines 397, 405, 410, 421, 471, captions of Figs 3, 5, 6, 7 & 8)

- line 389: "droplets with higher charging rate" suggests that within one simulation, different droplets have different charging rate, which is not the case, right?

- lines 406-407: four significant digits for percentage change grossly contrasts with the idealised setup of the simulations, with relatively small ensemble size, and with the (chaotic) nature of the modelled system

- line 494: check sentence grammar

- caption of Fig. 5: should be "water path", not "water precipitation" (three times)

**References**

- Forbes & Clark is cited but not listed in references

- Zhang et al. 2019 cited in line 38 but not listed in references (likely meant to be Zhang et al. 2018 which is listed but not referenced in the text)

- Frederick et al. 2018 should be Frederick and Tinsley 2018 in line 42

- Beard (2004) should be Beard et al. (2004) in line 48

- Tripathi et al. (2006) is cited (in line 58) but not listed

- Andronache et al. (2004) should likely be Andronache (2004) in line 77

- Köhler et al. 1936 should be Köhler 1936 in line 122

- Bott 1998 is cited in line 152 but not listed in the references

- Seeßlberg should be Seeßelberg in line 152

- Seinfeld and Pandis is cited in line 155 but not listed in references

- Fuchs 1964 is cited in line 156 but not listed in references

- Shima et al. 2014 is missing a permalink: http://hdl.handle.net/2115/55063 (it would also be worth indicating that it is in Japanese)

- Davis 1964a and Davis 1964b are cited but not included in the references (lines 208 and 210)

- "Zhang,2023" reference given in line 507 is not listed in the references

- Rogers & Yau is cited but Yau & Rogers is listed

- Pruppacher & Klett is cited but Pruppacher, Klett and Springer is listed (BTW, the DOI is 10.1007/978-0-306-48100-0, please also double check the year)

- Sato et al. 2017, 2018 and VanZanten et al. 2011 have doubled journal names in the reference list

- Sate et al. references are not listed in chronological order

- VanZanten vs. van Zanten

- Lasher-Trapp vs. Lasher-trapp

- Tinsley & Zhou 2006 is listed but not cited

- some journal names are abbreviated, some are not

- some surnames are in ALL-CAPS, some not

- some journal names are ALL-CAPS

- some titles are typeset with All Words in Caps, some with just the first word capitalised

- if citing a work with parentheses, avoid double "))" - lines: 86, 213, 500

---

## Author Comment (AC1)

Response to RC1 of manuscript egusphere-2023-2507 submitted to Geosci. Model. Dev.

Apr, 2024

**General comments:**

This study investigates the impact of enhanced coalescence due to droplet charge in high-fidelity simulations using the superdroplet method. While the methodology and results are thoroughly presented, the authors' key assumption of instantaneous droplet charging is not well-founded or discussed, and likely leads to a strong overestimate of the impact of droplet charge on precipitation. Similarly, the authors do not attempt to disentangle purely microphysical effects from flow field variability that stems from microphysics-flow coupling (vs. using a method like piggybacking). These reservations about the scientific merits combined with more minor concerns about the writing itself (including typos and inappropriate references) lead me to recommend major revisions before this paper be considered for publication.

We greatly appreciate the invaluable and constructive feedback provided by Reviewer #1. We have acted upon all the points raised. We believe the current manuscript is greatly improved through addressing the review comments and further elaborating the methods and results.

**Specific comments:**

1. The selected references for describing the importance of droplet coalescence (e.g. line 35) are not appropriate general references for this statement. For instance, Rosenfeld 2008 specifically concerns the a controversial mechanism of aerosol convective invigoration, which has relatively little to do with coalescence. Craig 1995 discusses radiative effects which are also not inherently specific to coalescence. Forbes & Clark does not even appear in the references. More appropriate citations would include review papers, chapters from the IPCC or a classical cloud microphysics textbook, or studies which specifically investigate droplet coalescence. Likewise in line 483: the stated impact of increasing aerosol concentration is the Twomey effect, and should not be attributed to Rosenfeld 2008!

   Thank you for your important comments and suggestions, we agree that the reference to droplet coalescence is not appropriate for the point we discussed. we modified the reference for describing the importance of droplet coalescence in line 35 and line 483 as follows:

   'Droplet coalescence is one of the main processes leading to precipitation and even cloud chemistry, affecting cloud microphysics and thereby changing the global radiation budget (Pruppacher and Klett, 2010, Chapter15; Grabowski and Wang, 2013; IPCC AR6 WG1 Ch7, 2021).'

   'An increase in the aerosol concentration decreases the effective radius by increasing the concentration of small droplets, which could have a significant impact on cloud formation (Twomey, 1974)'

2. I have some issues with your notation and definitions in section 2.4. Line 144: Given that both E0 and Ees are presumably efficiencies varying between 0 and 1, they should be multiplicative rather than additive. I am also confused by the notation of Rp, rp, QR, and qr, as the text describes these radii and charges as being general to "large droplets" or "small droplets", whereas I would imagine them to be descriptive of the larger droplet R and smaller droplet r for a given pair (R, r). What does the subscript "p" refer to?

   Thanks for your comments. We will answer your comments point by point:

*Re the presumed additive efficiencies: We incorporated both electro-coalescence efficiency and general coalescence efficiency using the approach outlined by Andronache 2004 (http://dx.doi.org/10.1016/j.jaerosci.2004.07.005 ). This method is the general way to handle electro-coalescence efficiency.*

*Re the notation of $R_p$, $r_p$, $Q_R$, and $q_r$: Sorry for misleading symbol, we use subscript "p" represent "particle", but as you pointed out it's confusing, we remove subscript "p" and replace "large droplets" or "small droplets" with "larger droplets of droplets pair" & "smaller droplets of droplets pair" in the text.*

3. The assumption in 212-213 that droplets charge instantaneously following coalescence seems like a major flaw in this study which would lead to a strong overestimate of the effects of electro-coalescence in an LES. Given that you are using the superdroplet method for this study, you should in fact be able to model the time response of charging on a given superdroplet as an additional attribute! This would provide a much more trustworthy study of the effects of droplet charge on coalescence that could actually be used to quantify and suggest whether this effect is notable. As it stands however, this assumption undermines the findings of this study and is not adequately discussed as a limitation or confounding factor in the abstract or conclusions. Furthermore, the values chosen for alpha in the numerica l experiments are not well-justified with values or ranges measured in real clouds.

*Thank you for your important comment. The droplets get charged instantaneously is a main assumption of this work, we make this assumption base on following two reasons:*

- *Based on the findings of Zhou et al. 2012, the charging time for 10-micrometer droplets is approximately ten minutes, which is relatively short compared to the overall development and precipitation duration of clouds. Moreover, for the droplets in our simulations, we have set a maximum limit of 50 elementary charges for droplets of any radius to constrain the impact of electro-coalescence effects from exceeding our estimates. Notably, 50 elementary charges are significantly less than the maximum charge that droplets can carry; observational data from Beard et al. 2004 reveal that real stratocumulus droplets of 10 micrometers carry about 100 elementary charges. Therefore, our assumption of instantaneous charging is based on realistic observations and numerical simulation results.*

- *As the reviewer suggested, we could simulate the charge amount and charging time dynamically as attributes of a super-droplet. However, because the electric potential gradient within clouds and the charge carried by droplets interact, the droplet's charge is not solely a function of time. As the charge attribute of a super-droplet changes at each time step, the cloud's electric potential gradient would also change, thereby affecting the charging efficiency. Addressing this would require significantly more computational resources. Currently, our parametrization is an initial attempt, and we aim to use simpler assumptions to model the electro-coalescence effect and obtain preliminary results. In future work, we also plan to parameterize the prediction of droplet charge.*

4. I also take issue with the comparisons made between different simulations given that the flow-field and microphysics are fully coupled. A more appropriate way to analyze the purely microphysical effects of electro-coalescence would be through the common technique of piggybacking, as other studies have shown that differences due to small perturbations to the flow field often outpace differences related to microphysics effects. I do like the approach of using 50 ensemble members to analyze statistics of the superdroplet simulations, but I'm not convinced that this would help isolate

microphysics from flow-field variability. Furthermore, in section 2.5 and 2.6, is it not clear whether the simulations performed are DNS or LES as there is no sub-grid scale turbulence model (line 340), nor is there any mention of what impact neglecting SGS turbulence would have on the results.

Thank you for your insightful comments. Regarding the disentanglement of purely microphysical effects from the flow field, we recognize the efficiency of the piggybacking technique in isolating microphysical impacts from dynamic interactions, as demonstrated in other studies. In this study, we constructed idealized simulations specifically designed to highlight the electro-coalescence effects on warm cumulus clouds. The results indeed showed a significant difference between the scenarios with and without electro-coalescence under identical flow conditions, emphasizing the distinct influence of electro-coalescence. For future work, we plan to incorporate the piggybacking technique to further isolate and verify the microphysical effects from flow-field variability. This will enhance our understanding and ensure a more robust analysis of the interplay between microphysics and atmospheric dynamics.

Re the simulations performed as DNS or LES, we are using SGS for turbulence model for dynamics, not using any SGS turbulence model for cloud microphysics, such as collision-coalescence enhancement, velocity fluctuation, and supersaturation fluctuation, the simulations performed is LES. We clarify the simulations performed is LES and the impact of neglecting sub-grid scale (SGS) turbulence as follows in line 360:

'We are employing a subgrid-scale (SGS) turbulence model for dynamic processes, but not for cloud microphysics processes such as collision-coalescence enhancement, velocity fluctuations, and supersaturation fluctuations. This approach may lead to an underestimation of the collision rate of charged droplets (Lu & Shaw 2015). The simulations conducted are based on the Large Eddy Simulation (LES) methodology.'

5.  In general when discussing figures and results, details from the figure caption which describe the various lines are repeated in the full text unnecessarily (e.g. lines 240-245, line 356-364, and elsewhere throughout). This repetition should be removed and avoided.

Thank you for your comment, we removed the repeat caption of plots in the text and rephase this sentence for each figure such as:

'Figure 1 displays a comparison of the collision-coalescence kernel for droplet radii of 40 μm (black lines), 20 μm (green lines), and 10 μm (red lines) across different calculation methods. The plots vary by line style to represent different analytical treatments and the inclusion or absence of static electric forces, with specific settings for the droplet charging rate shown in Figures 1(a) and 1(b).'

6.  Lines 401-406 suggest that the trend going from LA to MA conditions is opposite from the trend going from MA to HA conditions, when the figure in fact indicates that the trend is consistent. Increasing the aerosol concentration appears to uniformly delay and reduce the precipitation quantity.

Sorry for the inconsistent description and figures, we rephase the conclusion about precipitation on different aerosol background in lines 409-416 as follow:

'Under NC settings, the Twomey effect demonstrates that higher aerosol concentrations lead to smaller particle radii in clouds, reducing precipitation efficiency. Conversely, when electrostatic forces are introduced, these higher aerosol concentrations substantially enhance precipitation across different scenarios. Specifically, in high aerosol (HA) conditions, the precipitation enhancement reaches 782% over the no charge (NC) setting; for medium aerosol (MA) conditions, it's 467%

higher; and for low aerosol (LA) conditions, the increase is 110%. This illustrates the significant role electrostatic forces play in modulating cloud dynamics and precipitation responses to aerosol variations.'

**Technical comments**

- Line 33: "play a key role in cloud formation" should be "rain formation"

  Thanks for the comment, we replaced "cloud formation" with "rain formation" in line 27.

- Line 62: the Greenfield Gap should be concisely defined

  Thanks for your suggestion, we added a concisely definition of Greenfield Gap in lines 69-71:

  'The so-called Greenfield gap, identified by Greenfield (1957), describes the reduced concentrations of particles in the 0.1 to 1 micrometer size range. Greenfield gap could be eliminated with sufficient charging of the droplets.'

- Line 200: there is an extra close-parentheses ")" after Khain04

  Thank you for your comment, removed extra ")" in line200.

- Line 203: there is an extraneous comma after "the IM treatment"

  Thank you for your comment, removed extra "," in line203.

- Line 228: if alpha is a ratio, it should be unitless and have a maximum value of 1. Can you clarify the definition here?

  Thank you for your comment, the alpha is an empirical parameter, when alpha=2 for average conditions of strongly electrified clouds and has an upper limit of alpha=7 that can occur by conduction charging (Pruppacher & Klett, 1997, Chapter 18, based on available observations). we rephase the alpha definition part(Lines 242-244) as follow:

  'The charging rate $\alpha$ is an empirical parameter ($\alpha$ is referred to herein as the droplet charging rate) that varies between 0, which represents neutral particles, and 7, which represents highly electrified clouds associated with thunderstorms (Andronache, 2004).'

- Line 305-306: Why are the number concentrations written this way, as "3 x" something?

  Thanks for your comment, we agree that this statement is unreadable, and we have rephrased the sentence defining aerosol number concentration in lines 312-315 as follows:

  'The aerosol number concentration and size distribution were adjusted to 3, 6 or 9 times from that given in Van Zanten et al. (2011) for RICO intercomparison case. The aerosol number concentration and size distribution is given by a bimodal log-normal distribution: The particle number concentrations of the two modes are $N_1 = 90 \text{ cm}^{-3}$ and $N_2 = 15 \text{ cm}^{-3}$, respectively.'

- Line 312-313: "SDM requires less computational cost..." compared to what? Be specific; it requires more computational cost than a bulk method!

  Thanks for comment, we clarified SDM requires less computational cost than a bin model in line 325-326 as follow:

  'SDM requires less computational cost to accurately simulate clouds and precipitation compare to bin scheme (Shima et al., 2009)'

- Equation 27: what is p?

  Thanks for the comment, we remove the useless subscript 'p' in the formula.

- Lines 433-435 are repeated from earlier in the paragraph.

  Thanks for comment, removed repeat sentence in line436.

- Lines 485-490 are a very good summary of key findings and insights from this study

Thank you so much for your encouragement!

---

## Author Comment (AC2)

**1. General Response:**

The 'Preliminary evaluation of the effect of electro-coalescence with conducting sphere approximation on the formation of warm cumulus clouds using SCALE-SDM version 0.2.5-2.3.0' focuses on numerical modelling of particle coagulation in a Cumulus cloud typical of fair-weather convection. The focus is on studying the effects of droplet charges on the effectiveness of coagulation, and the resultant changes in rainfall properties. The study employs a two-dimensional idealised fluid dynamics setup resolved on a grid with 50m spacing, with no subgrid-scale dynamics representation. The particulate phase is represented with simulation point particles, each representing a sample of modelled droplets. The simulation particles undergo collisions (only with other simulation particles, not within the subpopulation represented by each of them), and the resultant coagulation efficiency is parameterised taking into account theoretical considerations for the enhancement of collisions probability due to the particles being charged. The simulations do not involve tracking particle charges - these are a priori determined as a function of the droplet size. Particle collisions are treated employing coagulation kernel approach, without assessing the inter-particle distances. Besides collisions, the particles are subject to condensational growth and evaporation, sedimentation and advection with the flow. The representation of collisions and the initial sampling of the simulation particle population is probabilistic, and the study analyses 50-member ensembles for each setup.

Unfortunately, it is hard not to begin with pointing out that the lax approach to text and figure composition, reference consistency and equation typesetting distracts from the paper's content and tarnishes overall impression. The manuscript was submitted prematurely and calls for a major revision.

We greatly appreciate the invaluable and constructive feedback provided by Reviewer #2. We have acted upon all the points raised. We believe the current manuscript is greatly improved through addressing the review comments and further elaborating the methods and results.

**2. Detailed Responses:**

**Overall impression and suggestion of a major revision**

Thank you for your comments, we will answer these comments and explain my revision point by point.

- **The choice of the particle-resolved method is not explained - what are the benefits, tradeoffs, limitations as compared to other modelling techniques, in the very context of modelling charged-particle interactions?**

  We appreciate this observation and have revised the explain the advantages and shortcomings of particle-base method compare to bin method and bulk method at the beginning of section **2. Description of the cloud model** (page: 3, line: 97-101) as follow:

  The particle-based microphysics method, which calculates the electro collision-coalescence kernel in real time, offers more detailed insights into droplet behavior influenced by electrostatic forces, surpassing the bin method that relies on lookup tables (Khain et al., 2004), while also demanding less computational resources.

- **What are the implications of one of the key assumptions, namely that the drop charge is merely parameterized as a function if its size?**

Thank you for pointing out this issue. Based on the formulas for the voltage near a charged spherical particle and the breakup voltage in air, we have expanded section **2.4 Collision-coalescence and the Electric Effect** (p7, l227-239), to include a detailed derivation of how the charge amount can be parameterized as a function of droplet radius:

The voltage near a charged spherical particle is described by $U = q / 4\pi\varepsilon_0 r^2$ (Bleaney and Bleaney 1993), here $\varepsilon_0 = 8.854\times10^{-12}$ is the dielectric permittivity of free space. The maximum charge that cloud droplets can carry is determined by the air breakdown voltage $U_b \sim 3\times10^6$ $Vm^{-1}$ (Meek and Craggs 1953). Consequently, the maximum charge that droplets can carry is as follows:

$$q_{max} = 4\pi U_b \varepsilon_0 r^2$$

(17)

To simulate droplets in weakly electric field, we following Andronache (2004), described the mean charges on the larger droplet and smaller droplet in a pair of droplets as a function of the droplet radii as follows:

$$Q_R = 4\alpha A R^2, q_r = 4\alpha A r^2 \tag{18}$$

Here $A = \pi \times U_b \times \varepsilon_0 \times 10^{-2} = 0.83\times10^{-6}$ is two orders of magnitude smaller than the maximum particle charge, representing weakly charge condition, and the charging rate $\alpha$ is an empirical parameter ($\alpha$ is referred to herein as the droplet charging rate, equal to ratio of particle charge amount and maximum possible charge) that varies between 0, which represents neutral particles, and 7, which represents highly electrified clouds associated with thunderstorms (Andronache, 2004).

- **Unlike the velocity-differential driven gravitational coagulation, the Brownian mode applies to same-sized particles, and thus should occur also within particles represented by a single simulation particle, which IIUC is not the case in the model?**

    Thank you for your comment. Brownian motion would not significantly affect the coalescence or collision of droplets of similar sizes. Brownian motion primarily influences the movement of very small particles, such as aerosols or dust smaller than 0.1 micrometers, rather than the larger droplets commonly found in clouds. Particles larger than 1 micrometer are predominantly governed by gravitational effects and fluid dynamics.

    For droplets of same size, the impact of Brownian motion is negligible. These droplets possess enough mass to render the random molecular collisions characteristic of Brownian motion inconsequential. Instead, their movement and interactions are more likely to be driven by factors like fluid dynamics, gravitational settling, and potentially electrical charges, rather than by Brownian motion.

    It's also worth noting that particles within the 0.1 to 1 micrometer range typically exhibit lower concentrations, a phenomenon known as the Greenfield gap. A key focus of our study is on how the electro-coalescence effect enhances the collision-coalescence kernel within this size range that increases rain in the warm cumulus cloud.

- **Despite the whole paper focuses on fair-weather convective cloud simulation, and only the**

**very last sentence of the Discussion section relates to fog, the Conclusions section puts forward a hypothesis that "Electro-coalescence has a larger impact on highly polluted warm clouds or fog" - are there grounds for this statement**

**in this study?**

Thank you for your observation. We removed the part that involved fog elimination in the **Discussion** and **Conclusion** section.

- **More background on electro-coalescence would help to cater to readers not acquainted with the earlier works of the authors, and to highlight the importance of the results by explaining a broader perspective on the challenges in this domain, from measurement, theoretical and numerical modelling perspectives.**

  Thank you for your comment. We have revised the **Introduction** to highlight previous contributions from authors more prominently. Additionally, we introduce the measurement of the electric field within an actual cloud and include the progress of previous work on the parameterization of electro-coalescence modeling:

In weakly electrified clouds, the accumulation of space charges on droplets is controlled by the diffusion of atmospheric ions produced by the cosmic ray flux, and the concentration is dependent on the ratio of attachment and recombination and the downward ionosphere-earth current density (Jz). When the Jz penetrates the cloud, the gradients of the electric field at the cloud boundary could generate net positively charged droplets at the upper cloud boundary and net negatively charged droplets at the lower boundary (Zhou and Tinsley, 2007; Nicoll and Harrison, 2016). The observations of Beard et al. (2004) revealed that with a Jz of 1-6 pAm-2 in stratocumulus and altostratus clouds, a cloud droplet with radius of 10 microns can accept approximately 100 elemental charges, which is consistent with the theoretical calculations by Zhou and Tinsley (2007). In the cumulus, with vertical convection, the charged droplets at the boundaries can be mixed, droplets with opposite sign charged affect by electro-coalescence. The maximum charge on the droplets is determined by the air breakdown voltage for corona discharge (Meek and Carggs, 1953) and is a quadratic function of the droplet radius (Khain et al., 2004; Andronache, 2004).

Numerous studies have focused on parameterizing the microphysics of the electro-coalescence of particles, the challenge is to approximate calculate the electrostatic force between charged droplets. In the 1970s, the collision efficiency of oppositely charged droplets evaluated with a centered Coulomb force indicated that only in strongly electrified clouds can the charge on droplets significantly affect cloud droplet coagulation (Wang et al., 1978). The series of trajectory simulation work by Tinsley et al. (2001, 2006), Zhou et al., (2009) and Tripathi et al., (2006) revealed that in a weakly electrified cloud, when taking into account the image charge force, the collision rate coefficient between the charged droplets could be different. Even with droplet charges of the same sign, the collision rate coefficient could be enhanced as a function of the charge on the particles with radii ranging from 0.1 microns to 10 microns (Zhou et al., 2009). Then, with a sufficiently large charge on the droplets, the so-called Greenfield gap (Greenfield, 1957) could be closed. Simulation results showed that for particles with radii smaller than 0.1 microns, when the particles obtain a large charge due to the evaporation of highly charged droplets, the collision rate coefficient is significantly decreased due to the repulsive electric force of droplets with charges of the same sign and is increased for charges of the opposite sign (Tinsley and Leddon, 2013). The updated simulation by Zhou et al. (2009) with an exact electric force treatment with the conducting sphere (CS) method indicated that the collision efficiency is a factor of two higher in the Greenfield

Gap than that from the results of single image charge (IM) treatment. A few laboratory experiment results were consistent with these theoretical simulations (Ardon-Dryer et al., 2015). These findings highlight the need to represent coagulation due to droplet and aerosol charges in the cloud model. Khain et al. (2004) (hereafter Khain04) conducted a 0-dimension simulation to study the effect of seeding charged droplets on a cumulus cloud using the spectral bin cloud model with a 4-dimensional (mass and charge rate of two droplets) collision efficiency lookup table based on the static electric force between charged droplets. The results showed a significant response in the evolution of clouds due to charged droplets. 5% of maximum charge amounts of natural droplets, which is 2.5 times larger than the results from Zhou et al. (2007), was used in Khain04 to investigate their influence on rain enhancement and fog elimination. Andronache (2004) and Wang et al. (2015) claimed that charged droplets significantly contribute to below cloud scavenging according to the analytical formula suggested by Davenport and Peters (1978), where the minimum amount of charge on droplets is 7% of the maximum limit. However, only Coulomb force (CB) treatment was used in Andronache's and Wang's simulation.

**Detail comments**

**Abstract:**
- **the first sentence of the abstract does not seem to apply to this paper - the analytic expression is not established here, it is used here? (if not, please clarify in the text)**

  Thank you for your comments, we apply the analytic expression of conducting sphere method to approximate the electrostatic force in this paper.
- **the abstract should clearly state what kind of simulations are done (2D, flow-coupled, capturing aerosol microphysics, warm rain, no subgrid dynamics, ...)**
- **it should be indicated that despite employment of a particle-resolved microphysics representation, the particle charges are not among the particle attributes**
- **similarly, worth clarifying that despite calculating drop trajectories, collisions are modelled assuming well-mixed coalescence volume assumption**
- **worth iterating the considered options of assumptions regarding charge treatment (e.g., charge polarity always opposite, ...)**

  Thank you for the comment, we clarified the simulation detail setup in the abstract as follow.

  This study employs 2D simulation in flow-coupled model that captures aerosol microphysics in warm cumulus case without relying on subgrid dynamics process, we assume droplets are always with opposite charge and well mixed in the cloud and charge is not a particle attribute in the simulation.
- **provision of numbers with 3 significant digits in the abstract (e.g., 5.42% higher) seems overzealous**

  Thank you for your comment. In response, we have simplified the abstract by removing three significant digits to more clearly present the conclusion.
- **it is worth to highlight the probabilistic nature of the simulations and the number of realisations employed in the study**

  Thank you for your comment. We have clarified in the abstract that each case involves 50 random runs for simulation realization as follows:

  To assess fluctuation effects, we conducted 50 simulations with varying pseudo-random number

sequences for each electro-coalescence treatment.

- **usage of the word "treatment" in the abstract suggests that CS is an alternative to super-droplet method (line 17)**

  Thank you for your observation, modified word "particle treatment" to "particle-based method" in abstract.

- **according to the GMD guidelines ([https://www.geoscientific-model-development.net/submission](https://www.geoscientific-model-development.net/submission).html#manuscriptcomposition), references should not be included in the abstract unless urgently required - suggest removing the reference to Khain 2004 (retaining the rest of the sentence)**

  Thank you for pointing out this issue. We have removed the reference from the abstract and apologize for the oversight in meeting the specifications.

**Code and data availability**

We agree with the comments of code and data availability, we have ==update the Zenodo code and data in [https://zenodo.org/records/11058066](https://zenodo.org/records/11058066)==.

- **the Zenodo archive contains four 9GB tar files without any annotation or metadata, with two-letter file names – one can guess that these contain simulation output, but provision of a description that can be accessed prior to downloading it would be helpful;**

  **Thank you for your kind comment,**

  Thanks for suggestions, we added description of the files in the Zenodo website:

  Ruyizhang2333/SCALE-SDM-electro-coalescence: SCALE-SDM-electro-coalescence-v0.3

  Ruyi Zhang

  The release used for Zhang et al. (2023).

  - Simulation codes and data analysis programs are provided in 'SCALE-SDM-electro-coalescence-v0.3.zip';
  - Package 'nc.tar' represents the simulation result of the 'No charge' setting;
  - Package 'cb.tar' represents the simulation result of the 'Columnb force' setting;
  - Package 'im.tar' represents the simulation result of the 'Image charge' setting;
  - Package 'cs.tar' represents the simulation result of the 'Conducting Sphere' setting;

  The user guide of SCALE-SDM please refers to 'README.md' within 'SCALE-SDM-electro-coalescence-v0.3.zip'

- **the referenced source code archive on Zenodo contains spurious non-portable compiler output (\*.o, \*.a, and \*.mod files) which should be removed**

  Thank you for the comment, we removed the compiler output files and update the release of model.

- **the title mentions v0.2.5-2.3.0 but the README.md file gives v0.0-2.2.2 - please make it consistent and add to Zenodo metadata**

  Thank you for the comment, we modified the model version within README.md file and update in Zenodo.

- **the "The data ... are available from the corresponding author upon request" statement is not in line with the journal policies - the statement implies lack of anonymous and persistent access to the data and should be removed, while clarification on what is included in the multi-gigabyte datafiles on Zenodo provided**

  Thanks for comment, we removed this sentence in the manuscript.

- **Trying to understand the contents of the provided contrib/SDM/sdm_coalescence.f90 file, I became puzzled with why all the electro-coalescence efficiency calculation lines (824, 825, 828, 830 and 834) are commented out, only to realise that the README.md file hints that such code blocks need to be manually uncommented if trying to reproduce the simulation results -**

**it seems to be quite an obscure way to provide the code. Please provide the source code in a way that no manual alterations are needed to reproduce the results and that the version identifier provided matches the unmodified code used for simulations.**

Thank you for your important comment and suggestion, in the latest version, we use two namelist var "sdm_elecol" and "sdm_elerate" to setup electro-coalescence scheme and charge rate in run script. After this modification, users could select electro-coalescence scheme and set charge rate in the run script without recompiling the model. We didn't modify the subroutine program of the electro-coalescence scheme and made a test run to make sure the simulation output was inconsistence with the previous version.

**Figures**

Thanks for the important comment and suggestion, ==we have modified the figures following all the comments.==

- **Fig 1: the kernel dimensionality should be volume over time, not volume over inverse-time**
- **the "*10-12" multiplication factor for Y axis in Fig. 1 is awkward given the 10-4 – 102 axis range – please avoid using scaling factors in labels**

  Thank you for the comment, we modified the unit of x axis of fig1.

[Figure]

[Figure]

Figure 1: Comparison of the effect of electric charge on the collision kernel for droplets sized 40 μm, 20 μm and 10 μm with small droplet radii between 10-2 μm and 10 μm. The charging rate α is 0.2 for panel (a) and 0.3 for panel (b). The solid line represents the results where the collision kernel is calculated by the analytical expression and treats the charged droplets as CS setting. The long-dashed line represents the results calculated by the analytical expression and treats the charge droplets as Khain04 setting. The dashed-dotted-dotted line represents the results calculated by the analytical expression and treats the electrostatic electric force by the IM setting. The dashed-dotted line represents the results calculated by the analytical expression and treats the electrostatic electric force by the CB setting. The dotted line shows the results with NC setting. The dashed line represents the results of the trajectory simulation according to Zhou et al. (2009).

- **typo in Fig 3b and 5d: "Liquid" not "Liquit"**

  Sorry for the misspell, modified fig3 x axis labels.

[Figure]

Figure 3: The time evolution of the domain-averaged precipitation amount (a) and the domain-averaged water path of the liquid water path (b), rainwater path (c) and cloud water path (d), which is consistent with Figure 2. The solid line represents NC setting, the dashed line represents CB setting, and the dashed-dotted line represents CS setting. The grey shade indicates the standard deviation calculated from 50 members of the random ensemble.

- **font sizes in the figure labels (incl. axis tick labels) should match text size, and be uniform, currently these span all sizes from unreadable small (Fig. 2!) to unreasonably large**
  Thank you for your comment, we replot the fig2 and make the title front size looks readable.

[Figure]

Figure 2: A comparison of the spatial structure of the mixing ratio of hydrometeors of the cumulus with NC setting, the electric force evaluated by CB setting and the electrostatic electric force evaluated by CS setting at times of 1500s, 2100s and 2700s. The charging rate α is 0.3.

- **the employment of a multiplication factor given above Y axis in figs 3a and 6-8 is misleading, non-standard and hard to notice, please use intuitive units instead**
- **labels are missing spaces between quantity name and unit parentheses, parentheses are sometimes ")()", sometimes "[]"**
- **the choice of X and Y axis ranges in figs 3a, 6-8 seems awkward (vast majority of the plot area is left blank for no good reason)**
- **some units are typeset in italics, some in normal font**
- **X axis in figs 6-8 would be more intuitive if presented in minutes/hours, while Y axis in mm**
- **figure quality is poor due to choice of inadequate raster graphics format, please use vector graphics**

Thank you for your careful and important comments and suggestions, we replot all figures as vector graphics, used intuitive units and axis range for fig3a and 6-8, made unit parentheses consistent and typeset in normal font, and made X axis in fig6-8 in hour and Y axis in mm.

**Equations**
- **eq. 27 - split into two equations - the comma is hard to interpret**

Thank you for your comment, we split eq.27 to two equations as follow.

$$\xi(a,x) = n(a,x,t=0)/(N_s(0)p) \qquad (28)$$

$$p(a,x) = p = \text{constant} \qquad (29)$$

- **parentheses in equations are typeset with misleading sizes (small parens surrounding large parens)**

  Thank you for pointing out this issue, e retyped all equations using the equation application 'MathType' to make the equations more specification.

- **using almost identical "a" and "α" right next to each other is very misleading (e.g., eq. 17)**

  Thank you for your comment, we modified 'a' to 'A' of eq.17 to make it more readable.

$$Q_R = 4\alpha A R^2, q_r = 4\alpha A r^2 \qquad (18)$$

- **suggest avoiding defining parameters with their units, e.g.: "α (cm-2)" and mentioning unitless values in the text – it is clearer to write "rate α ... with a range of 0.1 – 0.6 cm-2 ..."**

  Thank you for your comment, $\alpha$ is empirical parameter don't have unit, we remove this cm$^{-2}$ in the text.

**Text comments**

- **line 26: "electrostatics on charged droplets" seems awkward - please rephrase**

  Thanks for comment, we rephrase this sentence as follow:

  It is found that the electro-coalescence effect could affect cloud formation even when the droplet charge is lower charge limit.

- **line 37: cite Rayleigh 1879 (https://www.jstor.org/stable/113853)?**

- **line 114: likely worth referencing Montero-Mart´ınez et al. 2009 (https://doi.org/10.1029/2008GL037111)**

  Thanks for comment, we added cite Rayleigh 1879 and Montero-Mart´ınez et al. 2009 in the text.

- **line 62: suggest replacing well-known with so-called**

  Thanks for comment, we replaced 'well-known' with 'so-called'

- **line 72: 0-dimension but 4-dimensional**

  Thanks for your comment, this sentence is misleading, we rephrase this sentence as follow:

  Khain et al. (2004) (hereafter Khain04) conducted a 0-dimension simulation to study the effect of seeding charged droplets on a cumulus cloud using the spectral bin cloud model with a 4-dimensional (mass and charge rate of two droplets) collision efficiency lookup table based on the static electric force between charged droplets.

- **line 91: remove "accurate"?**

  Thanks for comment, we removed "accurate"

- **line 100: why the radius is "equivalent"? (+unintentional "::" at line end?)**

  Thank you for your comment, the use of an equivalent radius in the Super-Droplet Method (SDM), is fundamentally about simplifying the complex interactions and transformations of water droplets in clouds while maintaining accuracy in simulations. We removed the '::' at the line end.

- **line 104: replace "floating in the atmosphere" with "in the domain"?**

- **line 109: "sediments" -> "sedimentation"**

- **line 126: "heat conditions" -> "latent heat release"**

Thanks for comments, we replace these words as suggestions in the text.

- **line 131: mixing coefficients, units and functional dependencies in one expression leads to ambiguities - e.g., what does K means here - thermal conductivity (should be in italics) or Kelvin (all units should not be in italics, but with upright font)**

  Thanks for the comment, we rephase this sentence to make units in upright font.

- **line 134: the water density was already defined in line 111**

  Thanks for the comment, removed the repeat definition.

- **line 143: parenthesis in eq. (5) suggest at first sight that $\pi$ is a function and E is a constant, while the opposite holds - please try to format the equations intuitively**

  Thanks for the comment, we format the equation 5 as follow:

$$K = \pi E\left(\left(R+r\right)^2 \left|v_R^\infty - v_r^\infty\right|\right) + K_B \tag{5}$$

- **line 146: the single-particle size is given along with "mean" charge of particles - how the mean is defined, why isn't it a mean size as well? Should it be characteristic instead of mean? - please elaborate on the assumptions needed to define the notion of the kernel in four dimensional attribute space**

  Thanks for the comment, we call it 'mean charge' following Andronache 2004 (https://doi.org/10.1016/j.jaerosci.2004.07.005 ), 'mean charge' represent the statistically averaged representation of the charges carried by particles as a function of radius.

- **line 175: eq. (16) is referenced before being given**

  Thanks for the comment, removed eq (16) in line 175.

- **line 183: different symbol used for terminal velocity than before**

  Thanks for the comment, modified the symbol of terminal velocity to keep consistency.

- **line 184: collision -> colliding**
- **line 185: Fes -> Fes**
- **line 236: element -> elemental (also in lines 418 & 437)**

  Thanks for the comment, replace these words as suggestions.

- **line 257: "the interception effect" was never mentioned before**

  Thanks for the comment, we added explanation of the interception effect in line 272-274 as follow:

  The interception effect in particle collision coalescence refers to the process where smaller particles are captured by a larger droplet's boundary layer and swept into it, even without direct contact, due to the aerodynamic airflow around the falling droplet.

- **line 302: please rephrase "number and size distribution are made"**

  Thanks for comment, we rephase this sentence as follow:

  The aerosol number concentration and size distribution were adjusted to 3, 6 or 9 times from that given in Van Zanten et al. (2011) for RICO intercomparison case.

- **line 304: the composition was already given in line 301**

  Thanks for comment, removed repeat composition.

- **lines 305-307: subscripts, units with upright font; also: worth mentioning that these parameters are actually altered across simulations (section 3.3)**

  Thanks for comment, we modified units to upright font and note the aerosol parameter were altered in the simulation as follow:

  The aerosol number concentration and size distribution is given by a bimodal log-normal distribution: The particle number concentrations of the two modes are $N_1 = 90 \text{ cm}^{-3}$ and

$N_2 = 15$ cm$^{-3}$, respectively. Note that aerosol concentrations are multiplied by factors of 3, 6, or 9, depending on the aerosol background conditions.

- **line 310: it would be worth to clarify that Shima et al. (2009) includes warm-rain algorithm definition, while the 2020 paper includes mixed-phase extension (not used here) and coupling with SCALE (used here)**

  Thanks for comment, we pointed out we are running warm-rain simulation base on SCALE-SDM latest version as follow (line 332-335):

  This study concentrates on warm-rain microphysics. We developed a numerical simulation using the latest version of SCALE-SDM, specifically employing the SDM warm rain algorithm from Shima et al. (2009), rather than the SDM mixed-phase extension presented by Shima et al. (2020).

- **line 312: IIUC, the employed/developed SDM code is not available at the provided riken.jp URL – worth clarifying**

  Thanks for comment, we clarified the SDM code is not available in the riken url.

  Shima et al. (2009, 2020) constructed a particle-based cloud model SCALE-SDM by implementing the SDM into SCALE, which is a library of weather and climate models of the Earth and other planets (Nishizawa et al., 2015; Sato et al., 2015; https://scale.riken.jp). The SDM code is not accessible through this site.

- **line 313: provide reference for the lower cost mention**

  Thanks for comment, we added the reference mention lower computation cost (Shima et al., 2009).

- **line 330: this reads as if the fluid dynamics were not influenced by the latent heat budget of the particles, but surely are - should there be a third step defined?**

  Thanks for comment, we added third step as follow:

  The order of calculation in the model is as follows: 1) calculate the fluid dynamics without the coupling terms from the particles to moist air, and update the moist air; 2) update the super-droplets

  $\{\{\zeta_i, x_i, a_i\}\}$ from $t$ to $t + \Delta t$. 3) We integrate one cloud microphysics process one time step

  forward and then moves on to the next process.

- **line 333: I fail to understand the "Process lags in time is calculated preferentially" statement**

  Sorry for unclear statement, we rephrase this sentence as follow:

  Lagging processes are prioritized in computational priorities to ensure their synchronization and accurately describe their subsequent impact, consistent with overall system dynamics.

- **line 340: refer to Lu & Shaw 2015 (DNS study, https://doi.org/10.1063/1.4922645) and elaborate on the implications of lack of small-scale turbulence representation**

  Thanks for important suggestion, we added reference and elaborate on the implications of lack of small-scale turbulence representation as follow:

  We are employing a subgrid-scale (SGS) turbulence model for dynamic processes, but not for cloud microphysics processes such as collision-coalescence enhancement, velocity fluctuations, and supersaturation fluctuations. This approach may lead to an underestimation of the collision rate of charged droplets (Lu & Shaw 2015). The simulations conducted are based on the Large Eddy Simulation (LES) methodology.

- **line 351: please underline that this is just one of 50 realisations simulated, and given that and the fact that these plots are hardly distinguishable, are the conclusions really supported by this figure?**

Thanks for the comment, we clarified that fig2 from the single run and replot it, I agree that the first row plots (1500s) look similar, but as the rain develops, three setups are distinguishable and support the conclusions.

- **line 363: should be "standard deviation", not "standard deviation error", right?**

Sorry for the mistake, we plotted "standard error" here, modified all "standard deviation error" with 'standard error'.

- **lines 365-366: are three significant digits for a percentage increase meaningful here?**

Thanks for the comment, we think these three percentages better quantify the precipitation differences between the three methods.

- **line 388: is it both domain- and ensemble-averaged? (same remark for lines 397, 405, 410, 421, 471, captions of Figs 3, 5, 6, 7 & 8)**

Thanks for your comment, we modified all "domain averaged" with "domain and ensemble averaged" in the text.

- **line 389: "droplets with higher charging rate" suggests that within one simulation, different droplets have different charging rate, which is not the case, right?**

Sorry for the unclear statement, we rephase this sentence as follow:

However, cloud elimination is faster in higher charging rate condition, at 2700s, a lower charging rate condition results in a higher droplet mass density at a peak of approximately 1000 μm, which indicates that a higher charging rate results in a shorter lifetime of the cumulus cloud.

- **lines 406-407: four significant digits for percentage change grossly contrasts with the idealised setup of the simulations, with relatively small ensemble size, and with the (chaotic) nature of the modelled system**

Thank you for your comment. We conducted 50 ensemble simulations across three aerosol backgrounds, and the results support the notion that electro-coalescence increases precipitation in warm cumulus clouds. We believe these findings are convincing.

- **line 494: check sentence grammar**

Thanks for comment, we rephase this sentence as follow:

The electro-coalescence effect on a weakly electrified warm cumulus was revisited. Assuming droplets with opposite signs are charged instantaneously by $J_z$, the amount of charge is determined by the size of the droplets.

- **caption of Fig. 5: should be "water path", not "water precipitation" (three times)**

Thanks for the comment, we corrected the caption of fig5.

**References**
- **Forbes & Clark is cited but not listed in references**
- **Zhang et al. 2019 cited in line 38 but not listed in references (likely meant to be Zhang et al. 2018 which is listed but not referenced in the text)**
- **Frederick et al. 2018 should be Frederick and Tinsley 2018 in line 42**
- **Beard (2004) should be Beard et al. (2004) in line 48**
- **Tripathi et al. (2006) is cited (in line 58) but not listed**
- **Andronache et al. (2004) should likely be Andronache (2004) in line 77**
- **Köhler et al. 1936 should be Köhler 1936 in line 122**
- **Bott 1998 is cited in line 152 but not listed in the references**
- **Seeßlberg should be Seeßelberg in line 152**

- **Seinfeld and Pandis is cited in line 155 but not listed in references**
- **Fuchs 1964 is cited in line 156 but not listed in references**
- **Shima et al. 2014 is missing a permalink: http://hdl.handle.net/2115/55063 (it would also be worth indicating that it is in Japanese)**
- **Davis 1964a and Davis 1964b are cited but not included in the references (lines 208 and 210)**
- **"Zhang,2023" reference given in line 507 is not listed in the references**
- **Rogers & Yau is cited but Yau & Rogers is listed**
- **Pruppacher & Klett is cited but Pruppacher, Klett and Springer is listed (BTW, the DOI is 10.1007/978-0-306-48100-0, please also double check the year)**
- **Sato et al. 2017, 2018 and VanZanten et al. 2011 have doubled journal names in the reference list**
- **Sate et al. references are not listed in chronological order**
- **VanZanten vs. van Zanten**
- **Lasher-Trapp vs. Lasher-trapp**
- **Tinsley & Zhou 2006 is listed but not cited**
- **some journal names are abbreviated, some are not**
- **some surnames are in ALL-CAPS, some not**
- **some journal names are ALL-CAPS**
- **some titles are typeset with All Words in Caps, some with just the first word capitalised**
- **if citing a work with parentheses, avoid double "))" - lines: 86, 213, 500**

Thank you for your detailed technical feedback on the references part. We have carefully reviewed and addressed each point you listed, ensuring that all corrections and modifications have been implemented to meet the required standards. We appreciate your important review, which has undoubtedly enhanced the quality and accuracy of our manuscript.

---

## Referee Report (RR1)

I am pleased with many of the authors' replies to my concerns regarding the methods used in this paper, as well as their incorporation of suggestions regarding references, notation and language. Unfortunately, some of the justification present in the authors' response was not incorporated in the manuscript. I also have further concerns regarding remaining issues in the grammar and language throughout the work.

**Specific comments revisited:**

2. *Notations and E0 and Ees*: Thank you for updating the notation. The approach that you are following (Andronache) should be cited in the paper where you are defining the coalescence efficiency.

3. *Instantaneous charging assumption*: I appreciate the thorough response justifying this assumption, but would like to see it explicitly and clearly stated in the paper as a limitation and your justification. It would likewise be nice to see the authors' expectation of what impact this assumption may have on the results discussed in the work.

4. *DNS vs LES:* Please also state in the text that this is a two-dimensional LES. (I realize it is already stated in the abstract)

**Additional general comments:**
- Can you justify or comment on the use of such complex equations for the particle charge (e.g. Eq 14, versus Eq 13) and whether this additional complexity (a) adds considerable computational burden; (b) whether it is justified in terms of the difference in results produced.
- It would also be nice to see a brief discussion of any new insights that have arisen from this study in comparison to Khain04.
- In general there are also several grammar mistakes that remain in this manuscript. I will address a few examples below, but I hope that the authors will undergo thorough proofreading.
- Section 2.5 is likely unnecessary to include, as the reader could easily look at references for the compressible nonhydrostatic equations. I would recommend instead pointing to a citation for the implementation used in SCALE.

**Other comments** (based on the track-changes document line numbers)
- Abstract: Consider adding an introductory sentence at the start to introduce what electro-coalescence is or its impacts, something like line 520-521 in the discussion section.
- L17: "... particle-based microphysics method: **the** super-droplet method..."
- L20: "...dynamic process. **W**e assume..."
- L39: "**rain** formation" (was not corrected from first review)
- L58-59: This sentence doesn't make sense, especially the added phrase.

- L74: What would it mean to "eliminate" the Greenfield Gap? Do you mean that there would no longer be fewer particles in this size range?
- L291-292: Can you comment on this range (0.1-10um) and its correspondence to the Greenfield Gap?
- L361-362 and L367-368 seem redundant
- L375 seems like an odd place for this statement. It would be better to put it in the data availability section and remove the url here.
- L450-451 is redundant with the figure caption.

---

## Referee Report (RR2)

Overall, the manuscript is improved compared to the original submission, however neither all of the key points raised in the first round of review, nor the readability issues were addressed in a fully satisfactory manner.

The manuscript still contains numerous technical flaws in punctuation, grammar, symbol and physical units mismatches. My first earlier comment was that "*the choice of the particle-resolved method is not explained - what are the benefits, tradeoffs, limitations as compared to other modelling techniques, in the very context of modelling charged-particle interactions*". The introduced change vaguely states that "*particle-based microphysics method, which calculates the electro collision-coalescence kernel in real time, offers more detailed insights into droplet behavior influenced by electrostatic forces, surpassing the bin method that relies on lookup tables (Khain et al., 2004), while also demanding less computational resources*". Computational demands are not explored in the present paper at all. Lookup tables are an implementation detail and can be used with both bin- and particle-resolved methods for speeding up evaluation of multi-dimensional formulae. On the other hand, particle-resolved models surpass bin-resolved models in the tractability of aerosol-cloud interactions, what is partly leveraged in the present study (while Khain et al. 2004 resorted to prescribed initial droplet size distributions). Similarly, the fidelity of representation of particle collisions is argued in literature (by the paper co-author!) to be superior in particle-resolved models (e.g., section 4.4. in Liu et al. 2023, `https://doi.org/10.1007/s00376-022-2077-3`). It would be worth to elaborate on it both in the Introduction as well as in the last paragraph of the Discussion section. As of now, sections Discussion and Conclusion do not refer to the particle-resolved methodology at all. It seems as all the discussion and conclusions apply equally well to bin methods - if so, worth highlighting.

Despite introducing changes to the model code and providing new source archive at Zenodo, the title and text still refers to the version number from the original submission - a change in version number is needed.

Despite authors' statement on provision of vector graphics in figures, provided pdf evidently contains raster graphics, here is how the Fig. 1 looks like if zoomed:

[Figure]

**Detailed comments:**

- l. 16: remove size-resolved (it is unclear what size refers to)

- l. 17: add "probabilistic" before "particle-based"

- l. 17-20: split into multiple sentences, suggest adding information on the processes represented in the microphysics model

- l. 27: rephrase "droplet charge is lower charge limit"

- l. 38: do these references support "and even cloud chemistry"?

- l. 39: missing space in "Chapter15"

- l. 42: is the non-chronological order of references intentional?

- l. 55: why is the μm unit typeset in different font?

- l. 57: rephrase "opposite sign charged affect by"

- l. 65: rephrase "series of trajectory simulation work by"

- l. 65: is the non-chronological reference order intentional?

- l. 71: "micrometer" used here, but "micron" elsewhere

- l. 85: "5% of maximum charge amounts of natural droplets" seems unclear, also perhaps better not to start a sentence with a digit

- l. 90: avoid using surnames as person indications, these should be used only as reference labels (also, plural "simulations"?)

- l. 92: what "real-time" refers to? is it different than in other cited studies? (it is elaborated on in l. 101, but still unclear why so central - super-particle method could also use a lookup table, it is just a way of speeding up evaluation of a multi-argument function...)

- l. 95-97 the "will be addressed in future work" statement seems awkward for an introductory section

- l. 103: super-droplet method was already mentioned, but the acronym is only defined here - move the definition to first occurrence

- l. 104: -droplets vs. -droplet, also since SDM was just defined, why not start using the acronym?

- l. 109: multiplicity was never mentioned before

- l. 113: specifying particular chemical composition seems misleading at the level of method description - there is nothing in the method that constraints it to ammonium sulphate!

- l. 126: remove "and"

- l. 135: non-chronological order of references; also: should be Rogers & Yau instead of Yau & Rogers

- l. 147: worth mentioning here that charge effects on the equilibrium saturation vapour pressure are neglected (see, e.g., Weon & Je 2010, https://doi.org/10.1063/1.3430007) - then at least this section would be somewhat justified in the paper

- l. 178: "viscosity rate" $\rightsquigarrow$ "dynamic viscosity"

- l. 187: is this what is meant: "droplet accepts less than 1 elemental charge"?

- l. 194: is $\mu_a$ the same as $\mu$ defined in line 177?

- l. 204: $r_{nt}$ is defined as dimensionless ratio in (12) but eq. (13) suggests it should have length dimensionality

- l. 209: ditto - $r_{nt}^2$ is added to dimensional $r^2$

- l. 210: period at line beginning, unopened parenthesis...

- l. 233: $\epsilon_0$ was already defined in l. 205

- l. 233-235: two consecutive sentences begin with almost the same phrase

- l. 237: grammar: "we following Andronache (2004)"

- l. 281: R symbol mismatch - previously used for particle radius

- l. 295: "and the time derivatives for condensation/evaporation" - predicate missing?

- l. 313-314: "size distribution were adjusted to 3, 6 or 9 times" sounds as if size parameters were adjusted

- l. 321: shouldn't this section go before 2.6?

- l. 324: "code is not accessible through this site" should better go into the preceding parenthesis

- l. 343: "common" ⤳ "coupling"?

- l. 341: worth rephrasing "processes for aerosol/cloud/precipitation particles are integrated separately" as it seems misleading - aerosol, cloud and precipitation particles are not treated separately

- l. 350 (again): please elaborate what are "Lagging processes" and "overall system dynamics"

- l. 350: "prioritized in computational priorities" - pleonasm

- l. 398: rephrase "The results of the domain and …" (domain- and ensemble-averaged?)

- l. 401: "cloud" ⤳ "clouds" (or otherwise "produces")

- l. 454: suggest removing "will be evaluated in our next paper"

- l. 459: "103" ⤳ "on the order of 100" ?

- l. 468: rephrase "two factors larger"

- l. 472: what "the cloud model" refers to?

- l. 489: "the effective radius" was never introduced, in general this sentence appears quite abruptly here (perhaps introduce subsections in section 4?)

- l. 505: acknowledge that alpha was arbitrarily prescribed here

- l. 506: remove "we leave them for the future work"

- l. 508: please elaborate on how it can be done and how this works brings us closer?

- l. 550: Davis 1964a - is it different from Davis 1964b, if so, add needed information

- l. 552: Davis 1964b - add DOI: https://doi.org/10.1093/qjmam/17.4.499

- l. 601: add permanent URL: https://www.jstor.org/stable/113853

- l. 608: "eulerian" ⤳ "Eulerian"; "lagrangian" ⤳ "Lagrangian"

- l. 657: it is Rogers & Yau, not Yau & Rogers (already pointed out in the first round of review)

---

## Editor Decision (ED1)

I am pleased with many of the authors' replies to my concerns regarding the methods used in this paper, as well as their incorporation of suggestions regarding references, notation and language. Unfortunately, some of the justification present in the authors' response was not incorporated in the manuscript. I also have further concerns regarding remaining issues in the grammar and language throughout the work.

**Specific comments revisited:**

2. *Notations and E0 and Ees*: Thank you for updating the notation. The approach that you are following (Andronache) should be cited in the paper where you are defining the coalescence efficiency.

3. *Instantaneous charging assumption*: I appreciate the thorough response justifying this assumption, but would like to see it explicitly and clearly stated in the paper as a limitation and your justification. It would likewise be nice to see the authors' expectation of what impact this assumption may have on the results discussed in the work.

4. *DNS vs LES:* Please also state in the text that this is a two-dimensional LES. (I realize it is already stated in the abstract)

**Additional general comments:**
- Can you justify or comment on the use of such complex equations for the particle charge (e.g. Eq 14, versus Eq 13) and whether this additional complexity (a) adds considerable computational burden; (b) whether it is justified in terms of the difference in results produced.
- It would also be nice to see a brief discussion of any new insights that have arisen from this study in comparison to Khain04.
- In general there are also several grammar mistakes that remain in this manuscript. I will address a few examples below, but I hope that the authors will undergo thorough proofreading.
- Section 2.5 is likely unnecessary to include, as the reader could easily look at references for the compressible nonhydrostatic equations. I would recommend instead pointing to a citation for the implementation used in SCALE.

**Other comments** (based on the track-changes document line numbers)
- Abstract: Consider adding an introductory sentence at the start to introduce what electro-coalescence is or its impacts, something like line 520-521 in the discussion section.
- L17: "... particle-based microphysics method: **the** super-droplet method..."
- L20: "...dynamic process. **W**e assume..."
- L39: "**rain** formation" (was not corrected from first review)
- L58-59: This sentence doesn't make sense, especially the added phrase.

- L74: What would it mean to "eliminate" the Greenfield Gap? Do you mean that there would no longer be fewer particles in this size range?
- L291-292: Can you comment on this range (0.1-10um) and its correspondence to the Greenfield Gap?
- L361-362 and L367-368 seem redundant
- L375 seems like an odd place for this statement. It would be better to put it in the data availability section and remove the url here.
- L450-451 is redundant with the figure caption.

Overall, the manuscript is improved compared to the original submission, however neither all of the key points raised in the first round of review, nor the readability issues were addressed in a fully satisfactory manner.

The manuscript still contains numerous technical flaws in punctuation, grammar, symbol and physical units mismatches. My first earlier comment was that "*the choice of the particle-resolved method is not explained - what are the benefits, tradeoffs, limitations as compared to other modelling techniques, in the very context of modelling charged-particle interactions*". The introduced change vaguely states that "*particle-based microphysics method, which calculates the electro collision-coalescence kernel in real time, offers more detailed insights into droplet behavior influenced by electrostatic forces, surpassing the bin method that relies on lookup tables (Khain et al., 2004), while also demanding less computational resources*". Computational demands are not explored in the present paper at all. Lookup tables are an implementation detail and can be used with both bin- and particle-resolved methods for speeding up evaluation of multi-dimensional formulae. On the other hand, particle-resolved models surpass bin-resolved models in the tractability of aerosol-cloud interactions, what is partly leveraged in the present study (while Khain et al. 2004 resorted to prescribed initial droplet size distributions). Similarly, the fidelity of representation of particle collisions is argued in literature (by the paper co-author!) to be superior in particle-resolved models (e.g., section 4.4. in Liu et al. 2023, `https://doi.org/10.1007/s00376-022-2077-3`). It would be worth to elaborate on it both in the Introduction as well as in the last paragraph of the Discussion section. As of now, sections Discussion and Conclusion do not refer to the particle-resolved methodology at all. It seems as all the discussion and conclusions apply equally well to bin methods - if so, worth highlighting.

Despite introducing changes to the model code and providing new source archive at Zenodo, the title and text still refers to the version number from the original submission - a change in version number is needed.

Despite authors' statement on provision of vector graphics in figures, provided pdf evidently contains raster graphics, here is how the Fig. 1 looks like if zoomed:

[Figure]

**Detailed comments:**

- l. 16: remove size-resolved (it is unclear what size refers to)

- l. 17: add "probabilistic" before "particle-based"

- l. 17-20: split into multiple sentences, suggest adding information on the processes represented in the microphysics model

- l. 27: rephrase "droplet charge is lower charge limit"

- l. 38: do these references support "and even cloud chemistry"?

- l. 39: missing space in "Chapter15"

- l. 42: is the non-chronological order of references intentional?

- l. 55: why is the μm unit typeset in different font?

- l. 57: rephrase "opposite sign charged affect by"

- l. 65: rephrase "series of trajectory simulation work by"

- l. 65: is the non-chronological reference order intentional?

- l. 71: "micrometer" used here, but "micron" elsewhere

- l. 85: "5% of maximum charge amounts of natural droplets" seems unclear, also perhaps better not to start a sentence with a digit

- l. 90: avoid using surnames as person indications, these should be used only as reference labels (also, plural "simulations"?)

- l. 92: what "real-time" refers to? is it different than in other cited studies? (it is elaborated on in l. 101, but still unclear why so central - super-particle method could also use a lookup table, it is just a way of speeding up evaluation of a multi-argument function...)

- l. 95-97 the "will be addressed in future work" statement seems awkward for an introductory section

- l. 103: super-droplet method was already mentioned, but the acronym is only defined here - move the definition to first occurrence

- l. 104: -droplets vs. -droplet, also since SDM was just defined, why not start using the acronym?

- l. 109: multiplicity was never mentioned before

- l. 113: specifying particular chemical composition seems misleading at the level of method description - there is nothing in the method that constraints it to ammonium sulphate!

- l. 126: remove "and"

- l. 135: non-chronological order of references; also: should be Rogers & Yau instead of Yau & Rogers

- l. 147: worth mentioning here that charge effects on the equilibrium saturation vapour pressure are neglected (see, e.g., Weon & Je 2010, https://doi.org/10.1063/1.3430007) - then at least this section would be somewhat justified in the paper

- l. 178: "viscosity rate" ⤳ "dynamic viscosity"

- l. 187: is this what is meant: "droplet accepts less than 1 elemental charge"?

- l. 194: is $\mu_a$ the same as $\mu$ defined in line 177?

- l. 204: $r_{nt}$ is defined as dimensionless ratio in (12) but eq. (13) suggests it should have length dimensionality

- l. 209: ditto - $r_{nt}^2$ is added to dimensional $r^2$

- l. 210: period at line beginning, unopened parenthesis...

- l. 233: $\epsilon_0$ was already defined in l. 205

- l. 233-235: two consecutive sentences begin with almost the same phrase

- l. 237: grammar: "we following Andronache (2004)"

- l. 281: R symbol mismatch - previously used for particle radius

- l. 295: "and the time derivatives for condensation/evaporation" - predicate missing?

- l. 313-314: "size distribution were adjusted to 3, 6 or 9 times" sounds as if size parameters were adjusted

- l. 321: shouldn't this section go before 2.6?

- l. 324: "code is not accessible through this site" should better go into the preceding parenthesis

- l. 343: "common" $\rightsquigarrow$ "coupling"?

- l. 341: worth rephrasing "processes for aerosol/cloud/precipitation particles are integrated separately" as it seems misleading - aerosol, cloud and precipitation particles are not treated separately

- l. 350 (again): please elaborate what are "Lagging processes" and "overall system dynamics"

- l. 350: "prioritized in computational priorities" - pleonasm

- l. 398: rephrase "The results of the domain and ..." (domain- and ensemble-averaged?)

- l. 401: "cloud" $\rightsquigarrow$ "clouds" (or otherwise "produces")

- l. 454: suggest removing "will be evaluated in our next paper"

- l. 459: "103" $\rightsquigarrow$ "on the order of 100" ?

- l. 468: rephrase "two factors larger"

- l. 472: what "the cloud model" refers to?

- l. 489: "the effective radius" was never introduced, in general this sentence appears quite abruptly here (perhaps introduce subsections in section 4?)

- l. 505: acknowledge that alpha was arbitrarily prescribed here

- l. 506: remove "we leave them for the future work"

- l. 508: please elaborate on how it can be done and how this works brings us closer?

- l. 550: Davis 1964a - is it different from Davis 1964b, if so, add needed information

- l. 552: Davis 1964b - add DOI: https://doi.org/10.1093/qjmam/17.4.499

- l. 601: add permanent URL: https://www.jstor.org/stable/113853

- l. 608: "eulerian" $\rightsquigarrow$ "Eulerian"; "lagrangian" $\rightsquigarrow$ "Lagrangian"

- l. 657: it is Rogers & Yau, not Yau & Rogers (already pointed out in the first round of review)

---

## Author Response (AR2)

**Reviewer 1:**

**General comments:**

I am pleased with many of the authors' replies to my concerns regarding the methods used in this paper, as well as their incorporation of suggestions regarding references, notation and language. Unfortunately, some of the justification present in the authors' response was not incorporated in the manuscript. I also have further concerns regarding remaining issues in the grammar and language throughout the work.

We greatly appreciate the invaluable feedback provided by Reviewer 1. We have acted upon all points raised and check the grammar and language throughout the manuscript. We believe the current manuscript is improved through addressing the review comments.

**Specific comments revisited:**

2. Notations and E0 and Ees: Thank you for updating the notation. The approach that you are following (Andronache) should be cited in the paper where you are defining the coalescence efficiency.

Thank you for the comment, we cite the Andronache (2004) at the line178 and empresses the coalescence efficiency parameterization refer to the paper as follow:

'Referring to Andronache (2004), we propose a parameterization of the collision efficiency…'

3. Instantaneous charging assumption: I appreciate the thorough response justifying this assumption, but would like to see it explicitly and clearly stated in the paper as a limitation and your justification. It would likewise be nice to see the authors' expectation of what impact this assumption may have on the results discussed in the work.

Thank you for the important comments and suggestions, we agree with your point about clarifying the assumption introduction. We added the limitation of the assumption and result from that might be caused by the assumption in section 2.4 lines 221-232 as follows:

'Zhou and Tinsley (2012) observed that droplets with a 10 μm radius achieve 70% of their charge in 680 seconds. However, following Andronache (2004) simplification of the complex charging process, we assume that the charge on droplets resulting from collision-coalescence reaches equilibrium instantaneously. We also consider an extreme scenario where the charge polarity of two colliding droplets is always opposite. The assumption of instantaneous charging might lead to an overestimation of the electro-coalescence effect.'

4. DNS vs LES: Please also state in the text that this is a two-dimensional LES. (I realize it is already stated in the abstract)

Thank you for your comments! We agree that it should be stated in the text that the simulations used a 2D LES model, and we have clarified this in Section 2.7 "Numerical

Setup and Scheme", lines 361-362 as follows:
'Our simulations use two-dimensional Large Eddy Simulation (LES) methodology.'

**Additional general comments:**
• Can you justify or comment on the use of such complex equations for the particle charge (e.g. Eq 14, versus Eq 13) and whether this additional complexity (a) adds considerable computational burden; (b) whether it is justified in terms of the difference in results produced.
Thank you for your comments. The comparison of four electrostatic force settings is a key highlight of our work.
(a) Re the Computational Burden:
The simulation times for the Coulomb force (CB), image charge (IM), and Khain 2004 (Khain04) methods in the warm cumulus case are similar. The conductive sphere (CS) method takes about 30% longer. However, this increased computational time is justified by its higher accuracy and broader applicability across various droplet sizes.
(b) Re the Justification of Different Results:
Zhou and Tinsley (2008, doi: 10.1029/2008JD011527) demonstrate that the CS method offers superior numerical stability and accuracy, particularly for droplets of similar size. This enhanced precision is essential for accurately simulating cloud microphysical processes, making the additional computational burden worthwhile.
In conclusion, while the CS method incurs a higher computational burden, it provides superior numerical stability and accuracy. We also added a brief discussion on discussion section (lines 474-475) as follows:
'The CS method for electrostatic force should be incorporated into the cloud model, despite a 30% increase in computation time. CS method provides superior numerical stability and accuracy in simulating charge droplet interactions, particularly for charge droplets of similar size.'

• It would also be nice to see a brief discussion of any new insights that have arisen from this study in comparison to Khain04.
Thank you for your suggestion. We added a brief discussion in the discussion section to highlight the new insights gained from this study in comparison to Khain04. This will provide a clearer understanding of the advancements and differences between the two studies.
'Khain et al. (2004) evaluated electro-coalescence at a low charging rate of 5% of the maximum charge on droplets. In our simulation, we tested charging rates ($\alpha$) ranging from 0.05 to 0.6, equivalent to 0.15% to 1.8% of the maximum charge. At a charging rate of 0.3, the electric force evaluated by the CS method increased domain and ensemble-averaged precipitation by approximately 5.42% compared to the Khain04 setting. The results indicate that even with weak charging, the electro-coalescence effect significantly increases precipitation.'

• In general there are also several grammar mistakes that remain in this manuscript. I will address a few examples below, but I hope that the authors will undergo thorough

proofreading.

Thank you for pointing out the grammatical issues in our manuscript. We appreciate your specific examples and will undertake a thorough proofreading to address all grammatical errors.

• Section 2.5 is likely unnecessary to include, as the reader could easily look at references for the compressible nonhydrostatic equations. I would recommend instead pointing to a citation for the implementation used in SCALE.

Thank you for your comment. We deleted section 2.5, and cite Shima et al. 2020 in section numerical setup (Lines 304-307) as follow:

'The moist air fluid dynamics in this study are computed using eqs. (71)-(81) from Shima et al. (2020). The calculations utilize SCALE's dynamical core, employing a forward temporal integration scheme. This approach is implemented on an Arakawa-C staggered grid (Arakawa and Lamb, 1977) using a finite volume method.'

**Other comments** (based on the track-changes document line numbers)
• Abstract: Consider adding an introductory sentence at the start to introduce what electro-coalescence is or its impacts, something like line 520-521 in the discussion section.

Thanks for the important suggestion, we added a sentence to briefly introduce the electro-coalescence effect at the beginning of abstract as follows:

'The phenomenon electric fields applied to droplet induce droplet coalescence was called electro-coalescence effect.'

• L17: "… particle-based microphysics method: the super-droplet method…"

Thanks for comment. We rephrase this sentence into:

'To investigate this effect, we applied a weak electric field to a cumulus cloud using a size-resolved cloud model that employs the super-droplet method.'

• L20: "…dynamic process. We assume…"

Thanks for the comment. We rephrase the sentences to make them flow:

'… dynamics. In the simulation, we assume…'

• L39: "rain formation" (was not corrected from first review)

Sorry for our mistake. We modified 'cloud formation' to 'rain formation' in Line30.

• L58-59: This sentence doesn't make sense, especially the added phrase.

Thanks for the comment. we rephrase this sentence to introduce how electro-coalescence happened.

'In cumulus clouds, vertical convection causes positive charge droplets from the upper boundary and negative charge droplets from the lower boundary to mix, leading to electro-coalescence.'

• L74: What would it mean to "eliminate" the Greenfield Gap? Do you mean that there

would no longer be fewer particles in this size range?

Thank you for your valuable feedback. By 'eliminate the Greenfield Gap', we intended to convey that the enhanced collision efficiencies due to the electro-coalescence effect would significantly mitigate the traditionally reduced scavenging rates for particles in this size range. However, we acknowledge that 'eliminate' may not be the most accurate term. Therefore, we have replaced it with 'reduce' at L75 to better reflect the intended meaning. Thank you for bringing this to our attention.

• L291-292: Can you comment on this range (0.1-10um) and its correspondence to the Greenfield Gap?

Thank you for your comment. We have added the connection between the 0.1-10 μm range and the Greenfield Gap as follows:

'The results indicate that the primary range for electro-coalescence is approximately 0.1 μm to 10 μm, which encompasses the Greenfield Gap.'

• L361-362 and L367-368 seem redundant
• L375 seems like an odd place for this statement. It would be better to put it in the data availability section and remove the url here.
• L450-451 is redundant with the figure caption.

Thanks for your important suggestions. We removed the redundant part and move the url to data availability section.

**Reviewer 2:**

**General comments:**

Overall, the manuscript is improved compared to the original submission, however neither all of the key points raised in the first round of review, nor the readability issues were addressed in a fully satisfactory manner.

We sincerely appreciate the invaluable and constructive feedback provided by Reviewer 2. We have addressed all the points raised and believe that the current manuscript has been significantly improved by incorporating the review comments, further elaborating on the benefits of SDM, and resolving the technical issues.

The manuscript still contains numerous technical flaws in punctuation, grammar, symbol and physical units mismatches. My first earlier comment was that "the choice of the particle-resolved method is not explained -what are the benefits, tradeoffs, limitations as compared to other modelling techniques, in the very context of modelling charged-particle interactions". The introduced change vaguely states that "particle-based microphysics method, which calculates the electro collision-coalescence kernel in real time, offers more detailed insights into droplet behavior influenced by electrostatic forces, surpassing the bin method that relies on lookup tables (Khain et al., 2004), while also demanding less computational resources". Computational demands are not explored in the present paper at all. Lookup tables are an implementation detail and can be used with both bin- and particle-resolved methods for speeding up evaluation of multi-dimensional formulae. On the other hand, particle-resolved models surpass bin-resolved models in the tractability of aerosol-cloud interactions, what is partly leveraged in the present study (while Khain et al. 2004 resorted to prescribed initial droplet size distributions). Similarly, the fidelity of representation of particle collisions is argued in literature (by the paper co-author!) to be superior in particle-resolved models (e.g., section 4.4. in Liu et al. 2023, https://doi.org/10.1007/s00376-022-2077-3). It would be worth to elaborate on it both in the Introduction as well as in the last paragraph of the Discussion section. As of now, sections Discussion and Conclusion do not refer to the particle-resolved methodology at all. It seems as all the discussion and conclusions apply equally well to bin methods - if so, worth highlighting.

Thank you for your thoughtful explanation and attached reference.

- Re the technical issues: We addressed and corrected them in the text and the Detailed comments section.

- Re the elaborate of SDM: We agree your point about elaboration the reason of choose particle-resolved method worth to mention in the Introduction, Discussion and Conclusion section. Sorry didn't make sense on last modification.

  In Introduction section, we elaborate SDM as follows:

  l.91-93: 'Lagrangian particle-based approaches accurate solutions for the collision-coalescence process compared to bin microphysics schemes, as they overcome the limitations imposed by the assumptions of bin schemes (Grabowski et al., 2019; Liu et al., 2023).'

  We rephrase the sentence in Description of the cloud model to clarify the benefit, the limitation and trade off of SDM compare to bin scheme, as follows:

l.101-107: 'In this study, we assume that the charged droplets are well-mixed in the warm cumulus cloud and focus on the electro-coalescence effect. A Lagrangian particle-based cloud model is used with the particle size resolved treatment following the SDM by Shima et al. (2009, 2020). Compared to bin microphysics schemes, the Super-Droplet Method (SDM) eliminates numerical diffusion and provides more accurate solutions for well-mixed volumes (Grabowski et al., 2019). Despite its sensitivity to super-droplet initialization and a higher variance than observed in reality (Liu et al., 2023), SDM is well-suited for this study.'

We added the superior of SDM in Discussion as follows:

l.467-469: 'The particle-based approach SDM provides explicitly cloud-aerosol interaction simulation, such as the role of CCN in rain formation (Grabowski et al., 2019). According to our simulation results, the electro-coalescence effect on precipitation is sensitive to the aerosol concentration.'

We also mention the importance of SDM in Conclusion as follows:

l. 479-481: 'A new simulation with the exact treatment of the electrostatic force for opposite sign charge case based on the particle-based approach SDM provides a good estimation of the effect of electro-coalescence in the Greenfield gap region.'

Despite introducing changes to the model code and providing new source archive at Zenodo, the title and text still refers to the version number from the original submission - a change in version number is needed.

Thank you for your comment. The changes made to the model code are optimizations in code writing and do not affect the microphysics processes or simulation results. Therefore, we believe we are still working with SDM version 2.3.0 and did not alter the version number in the text and title.

Despite authors' statement on provision of vector graphics in figures, provided pdf evidently contains raster graphics.

Thanks for the comment. We checked the graphics in manuscript and replaced the raster graphics to vector graphics. (e.g., zoom of fig.1:)

[Figure]

**Detailed comments:**

- l. 16: remove size-resolved (it is unclear what size refers to)
  Thanks for your comment. Removed 'size-resolved' in the sentence.
- l. 17: add "probabilistic" before "particle-based"
  Thanks for your comment. Added 'probabilistic' before 'particle-based'.
- l. 17-20: split into multiple sentences, suggest adding information on the processes represented in the microphysics model.
  Thanks for the suggestion. Split into three sentences and added microphysics processes in the model.
- l. 27: rephrase "droplet charge is lower charge limit"
  Thanks for the comment. rephrase 'droplet charge is lower charge limit' to 'droplet charge is at the lower charge limit'
- l. 38: do these references support "and even cloud chemistry"?
  Thank you for comment. These references do not support 'and even cloud chemistry', we removed this part in the text.
- l. 39: missing space in "Chapter15"
  Thanks for comment. Added space in 'Chapter 15'.
- l. 42: is the non-chronological order of references intentional?
  Thanks for your comment. Not intention, sorry about that…We modified these references in chronological order.
- l. 55: why is the µm unit typeset in different font?
  Thanks for your comment. It is modified to the same font.
- l. 57: rephrase "opposite sign charged affect by"
  Thanks for the comment. rephrase this sentence to:
  'In cumulus clouds, vertical convection causes positive charge droplets from the upper boundary and negative charge droplets from the lower boundary to mix, leading to electro-coalescence.'

- l. 65: rephrase "series of trajectory simulation work by"

  Thanks for the comment. rephrase this sentence to:

  'The trajectory simulation studies by'
- l. 65: is the non-chronological reference order intentional?

  Thank you for the comment. Rephrase the references in chronological order.
- l. 71: "micrometer" used here, but "micron" elsewhere

  Thanks for comment. Replace 'micron' and 'micrometer' by 'µm'.
- l. 85: "5% of maximum charge amounts of natural droplets" seems unclear, also perhaps better not to start a sentence with a digit

  Thanks for the comment. Rephrase the sentence to:

  'Khain04 set a charge rate equal to 5% of the maximum charge of natural droplets, which is 2.5 times larger than the values reported by Zhou et al. (2007), to study electro-coalescence impact on rain enhancement and fog elimination.'
- l. 90: avoid using surnames as person indications, these should be used only as reference labels (also, plural "simulations"?)

  Thanks for your comment. Modified surnames by references and plural:

  '…was used in Andronache (2004) and Wang et al. (2015) simulations.'
- l. 92: what "real-time" refers to? is it different than in other cited studies? (it is elaborated on in l. 101, but still unclear why so central - super-particle method could also use a lookup table, it is just a way of speeding up evaluation of a multi-argument function...)

  Thank you for your comment. We agree with your point and rephrase this sentence as follows:

  l.91-96: 'Lagrangian particle-based approaches accurate solutions for the collision-coalescence process compared to bin microphysics schemes, as they overcome the limitations imposed by the assumptions of bin schemes (Grabowski et al., 2019; Liu et al., 2023). In this study, we estimate the effect of electro-coalescence from $J_z$ on warm cumulus clouds by an exact treatment of electric forces using the conducting sphere (CS) method, using the Super-Droplet Method (SDM), a Lagrangian particle-based cloud microphysics scheme.'
- l. 95-97 the "will be addressed in future work" statement seems awkward for an introductory section

  Thanks for the comment. We removed this pert from the sentence.
- l. 103: super-droplet method was already mentioned, but the acronym is only defined here - move the definition to first occurrence

  Thanks for the comment. We moved the acronym to the introduction section (Line 95).
- l. 104: -droplets vs. -droplet, also since SDM was just defined, why not start using the acronym?

  Thanks for the suggestion. We replaced all the 'super-droplet method' by the acronym after first defined.
- l. 109: multiplicity was never mentioned before

  Thanks for the comment. the 'multiplicity' indicate the 'super-droplet represents multiple droplets with the same attributes and position' at the begging of the

sentence.

- l. 113: specifying particular chemical composition seems misleading at the level of method description -there is nothing in the method that constraints it to ammonium sulphate!

  Thanks for the comment. We removed 'ammonium sulphate' and rephrase the sentence.

- l. 126: remove "and"

  Thanks for comment. Removed 'and' by ','.

- l. 135: non-chronological order of references; also: should be Rogers & Yau instead of Yau & Rogers

  Thanks for the comment. Rephrase the references in chronological order and replaced 'Yau & Rogers' by 'Rogers & Yau'. Sorry for didn't correct this issue in the first round referee.

- l. 147: worth mentioning here that charge effects on the equilibrium saturation vapour pressure are neglected (see, e.g., Weon & Je 2010, https://doi.org/10.1063/1.3430007) - then at least this section would be somewhat justified in the paper

  Thanks for the comment and important suggestion. We mention charge effect on the equilibrium saturation vapour pressure as follows:

  'Note the charge-induced reduction in surface tension decreases the equilibrium vapor pressure (Weon & Je 2010).'

- l. 178: "viscosity rate" ~ "dynamic viscosity"

  Thanks for the comment. Replaced 'viscosity rate' by 'dynamic viscosity' at line 179.

- l. 187: is this what is meant: "droplet accepts less than 1 elemental charge"?

  Thanks for the comment. We rephrase this sentence to make it more clarify and readable as follows:

  'The rear collision range is relevant for droplets smaller than 0.1 µm, the droplets typically accept fewer than 1 elemental charge, meaning the electric force does not significantly impact the collision process.'

- l. 194: is $\mu_a$ the same as $\mu$ defined in line 177?

  Thanks for the comment. Yes, they are same, deleted repeat definition.

- l. 204: $r_{nt}$ is defined as dimensionless ratio in (12) but eq. (13) suggests it should have length dimensionality

- Thanks for your important comment. We mixed up two symbols for the distance of droplets. The symbols of eq (12) and (13) are corrected.

- l. 209: ditto - r2nt is added to dimensional r2

  Thanks for the comment. Ditto.

- l. 210: period at line beginning, unopened parenthesis...

  Thank you for your review. These technical issues have been corrected.

- l. 233: $\epsilon_0$ was already defined in l. 205

  Thanks for the comment. Deleted repeat definition.

- l. 233-235: two consecutive sentences begin with almost the same phrase

Thanks for the comment. Rephrase two sentences as follows:

'The air breakdown voltage: $U_b \sim 3 \times 10^6 \text{ Vm}^{-1}$ , determines the maximum charge that cloud droplets can carry (Meek and Craggs, 1953).   Consequently, the maximum charge that droplets can carry…'

- l. 237: grammar: "we following Andronache (2004)"

  Thanks for the comment. Rephrased this sentence as follows:

  'To simulate droplets in a weak electric field, we followed Andronache (2004) and described the mean charges on the larger and smaller droplets in a pair as a function of their radii as follows'

- l. 281: R symbol mismatch - previously used for particle radius

- l. 295: "and the time derivatives for condensation/evaporation" - predicate missing?

  Thanks for the comments. We deleted the initial section 2.5 followed the suggestion of reviewer 1.

- l. 313-314: "size distribution were adjusted to 3, 6 or 9 times" sounds as if size parameters were adjusted

  Thanks for the comment. We rephrase this sentence, split the data source and aerosol background factor into two sentences as follows:

  'The aerosol number concentration and size distribution were based on the data provided by Van Zanten et al. (2011) for the RICO intercomparison case. Note that aerosol concentrations are multiplied by factors of 3, 6, or 9, depending on the aerosol background conditions.'

- l. 321: shouldn't this section go before 2.6?

  Thanks for the comment. The 'Numerical setup and schemes' section is moved before 'Design of our numerical experiment' section.

- l. 324: "code is not accessible through this site" should better go into the preceding parenthesis

  Thanks for the comment. The url of SCALE and the sentence about accessible were moved to the 'Code and data availability' section to keep text coherence.

- l. 343: "common" ~ "coupling"?

  Thanks for comment. The 'common time step' refers to the uniform time interval at which the primary calculations of the SDM are updated, followed the jargon of section 5.4 'Operator splitting of the time integration' of Shima et al. (2020).

- l. 341: worth rephrasing "processes for aerosol/cloud/precipitation particles are integrated separately" as it seems misleading - aerosol, cloud and precipitation particles are not treated separately

  Thanks for comment. We removed 'for aerosol/cloud/precipitation particles' for disambiguation.

- l. 350 (again): please elaborate what are "Lagging processes" and "overall system dynamics"

  Thanks for the comment.

- Re the 'Lagging processes':

  Lagging processes refer to those processes that occur on shorter timescales within

the simulation, such as condensation and evaporation. These processes can change rapidly and thus require more frequent computational updates to ensure accurate representation. Prioritizing these processes in the computation helps to capture their dynamics effectively and prevents numerical instability.

- Re the 'overall system dynamics':
Overall system dynamics encompass the behavior and evolution of the entire system over time, considering the interactions and feedbacks between various components and processes. In our simulation, this includes the interactions between fluid dynamics, cloud microphysics.
To enhance readability and comprehension, we have rephrased the sentence as follows:
'Processes with shorter timescales are prioritized in the computation to ensure they accurately capture their subsequent impacts.'

- l. 350: "prioritized in computational priorities" – pleonasm
Thanks for the comment. Removed 'priorities'.

- l. 398: rephrase "The results of the domain and ..." (domain- and ensemble-averaged?)
Thanks for the comment. Rephrase the sentence beginning to :'The results of the domain water path, averaged over 50 ensembles…'

- l. 401: "cloud" ~ "clouds" (or otherwise "produces")

  Thanks for the comment. Modified 'cloud' to 'clouds'.

- l. 454: suggest removing "will be evaluated in our next paper"
Thanks for the comment. Removed 'and will be evaluated in our next paper'.

- l. 459: "103" ~ "on the order of 100" ?

  Thanks for the suggestion. replaced '103' by 'on the order of 100'.

- l. 468: rephrase "two factors larger"
Thanks for the comment. Rephrased to 'is twice as large as'

- l. 472: what "the cloud model" refers to?
Thanks for the comment. Replaced 'cloud model' to 'cloud microphysics scheme'.

- l. 489: "the effective radius" was never introduced, in general this sentence appears quite abruptly here (perhaps introduce subsections in section 4?)
Thank you for the comment. We agree that this sentence is abrupt and have removed it. We have retained the discussion of the simulation results.

- l. 505: acknowledge that alpha was arbitrarily prescribed here
Thanks for the comment. We have added the acknowledgment of alpha as follows '…as a function of the arbitrarily prescribed charging rate $\alpha$...'

- l. 506: remove "we leave them for the future work"
Thanks for the comment. We have removed 'and we leave them for the future work'.

- l. 508: please elaborate on how it can be done and how this works brings us closer?
Thanks for the comment. We elaborate on the outlook of this study in section conclusion as follows:

'Cloud radiation feedback is one of the sources of uncertainty in the climate model (Zelinka et al. 2017). The electrostatic force effect parameterization for different cloud types should be indicated to improve climate model accuracy. This study reveals the electrostatic force effect on warm cumulus clouds, contributing to the parameterization of electrostatic microphysical processes.'

- l. 550: Davis 1964a - is it different from Davis 1964b, if so, add needed information
  Thanks for the comment. We rechecked they are the same, deleted repeat references.
- l. 552: Davis 1964b - add DOI: https://doi.org/10.1093/qjmam/17.4.499
  Thanks for the comment. We have added the doi to reference of Davis 1964b.
- l. 601: add permanent URL: https://www.jstor.org/stable/113853
  Thanks for the comment and suggestion. Added url to reference of Rayleigh 1878.

- l. 608: "eulerian" ~ "Eulerian"; "lagrangian" ~ "Lagrangian"

  Thanks for the comment. We corrected the spell mistake at the reference of Sato et al. (2018).

l. 657: it is Rogers & Yau, not Yau & Rogers (already pointed out in the first round of review)

Thanks for the comment. We apologize for the oversight. We have corrected the reference to "Rogers & Yau" as suggested.

---

## Author Response (AR3)

Response to topic editor of manuscript egusphere-2023-2507 submitted to Geosci. Model. Dev.

Jul, 2024

**General comments:**

I support the manuscript for publication after several technical and grammar corrections are applied following GMD guidelines: https://www.geoscientific-model-development.net/submission.html

We greatly appreciate the invaluable feedback and help provided by topic editor. We have acted upon all points raised and check the grammar, units, citations and figures throughout the manuscript. We have also corrected two unclear sentences (lines 291 and 308). We believe the current manuscript is improved through addressing the review comments.

**Detail comments:**

- Spaces must be included between number and unit (e.g. 1 %). Please consider this at multiple instances within the text (Lines 34, 85, 89, 227, 253, 356, 370, 384, 385, 393, 394, 395, 405, 406, 407, 413, 448, 449, 452, 454, 456).

Thank you for your comment! We have corrected the format of the unit and ensured that spaces are included between the number and unit throughout the text.

- Units must be written exponentially (e.g. g m–3). Please consider this at multiple instances within the text (e.g., Lines 147, 180, ...) as well as on Figures and in Figure captions.

Thanks for the important comment. We have modified the units to exponentially throughout the manuscript.

- The abbreviation "Fig." should be used when it appears in running text and should be followed by a number unless it comes at the beginning of a sentence, e.g.: "The results are depicted in Fig. 5. Figure 9 reveals that...". Please consider this at multiple instances within the text (e.g., Lines 365, 376, 378, 382, ...)

Thanks for your comment and explanation. We have modified the abbreviation of figures in the running text accordingly.

- Line 134: "equation (2)" should be "Eq. (2)".

Thank you for the comment. We have modified the abbreviation of equation in line 134.

- Please make multiple changes regarding abbreviations:

a) I would advise you to omit definitions of several abbreviations in the Abstract (SDM, LES, NC), since you don't use them later in the Abstract. The abbreviations have to be newly defined in the Main text anyway.

Thanks for the comment. We have removed the redundant abbreviations in the Abstract.

b) The abbreviation "SDM" is defined twice within the Main text (Line 95, Line 104).

Thanks for the comment. We have removed the redundant definition of "SDM" in line

104.

c) The abbreviation "CS" is defined twice within the Main text (Line 76, Line 95).
Thanks for the comment. We removed the redundant definition of CS in l. 95.

d) The abbreviation "NC" is defined in Line 393; but it should be instead defined earlier (at the first instance in the Main text).
Thanks for the comment. We have defined the abbreviation "NC" in line 347 and removed the redundant definition in line 393.

e) The abbreviation "CB" is defined in Line 90; but it should be instead defined earlier (at the first instance in the Main text).
Thanks for the comment. We have defined the abbreviation "CB" in line 63 and removed the redundant definition in line 90.

- I can notice both "Greenfield Gap" and "Greenfield gap" appearing at multiple places within the manuscript. Please use consistent terminology.
Thanks for the comment. We have made the terminology consistent as "Greenfield gap" in lines 77 and 263.

- Please correct reference citations at multiple places within the text, for example:
a) Lines 98 and 484: "(Tinsley and Zhou 2015)" should be "(Tinsley and Zhou, 2015)";
b) Line 487: "(Zelinka et al. 2017)" should be "(Zelinka et al., 2017)";
...
Thanks for the comment. We have double-checked the citations in the text and corrected them accordingly (lines 47, 98, 129, 138, 151, 238, 484, 487).

- A few additional comments regarding optimization of the Figures:

a) As already previously pointed out by the reviewer: the choice of axis ranges in Figs. 3a, 6, 7, 8 is not optimal, since the vast majority of plot area is left blank.
Thanks for your comment. We have adjusted the axis ranges in Figs. 3a, 6, 7, and 8 to utilize the plot area more effectively like:

[Figure]

b) Figure 3: "IM setting" should also be mentioned in figure caption. I would further advise you to employ different colors for displaying results of various settings (NC, CB, CS, IM) in order to improve the readability of the figure (while you can retain different line types).

Thank you for the comment. We have added the "IM setting" information in the Figure 3 caption and changed the colors of different settings in Figures 3-8 to improve readability.

c) Figures 4, 5, 6, 7, 8: Similarly as in Fig. 3 I would advise you to use different (and contrasting ! ) colors for various results displayed on plot. For example, you often use grey and black lines on the same plot, which might be difficult to distinguish. Choosing bright yellow color as you do in Fig. 8 is also not optimal. Thus please improve the quality of figures.

Thanks for your suggestion. Ditto, we have modified the colors and axis ranges in the revised manuscript to improve the quality of the figures.